# Metabolic and Molecular Rearrangements of Sauvignon Blanc (*Vitis vinifera* L.) Berries in Response to Foliar Applications of Specific Dry Yeast

**DOI:** 10.3390/plants12193423

**Published:** 2023-09-28

**Authors:** Marta Rodrigues, Cristian Forestan, Laura Ravazzolo, Philippe Hugueney, Raymonde Baltenweck, Angela Rasori, Valerio Cardillo, Pietro Carraro, Mario Malagoli, Stefano Brizzolara, Silvia Quaggiotti, Duilio Porro, Franco Meggio, Claudio Bonghi, Fabrizio Battista, Benedetto Ruperti

**Affiliations:** 1Department of Agronomy, Food, Natural Resources, Animals and Environment (DAFNAE), University of Padova, 35020 Padova, Italy; marta.rodrigues@phd.unipd.it (M.R.); laura.ravazzolo@unipd.it (L.R.); angela.rasori@unipd.it (A.R.); valerio.cardillo@unipd.it (V.C.); pietro.carraro@phd.unipd.it (P.C.); mario.malagoli@unipd.it (M.M.); silvia.quaggiotti@unipd.it (S.Q.); franco.meggio@unipd.it (F.M.); claudio.bonghi@unipd.it (C.B.); 2Department of Agricultural and Food Sciences, University of Bologna, 40127 Bologna, Italy; cristian.forestan@unibo.it; 3National Research Institute for Agriculture, Food and Environment (INRAE), SVQV UMR A1131, University of Strasbourg, 67081 Strasbourg, France; philippe.hugueney@inrae.fr (P.H.); raymonde.baltenweck@inrae.fr (R.B.); 4Crop Science Research Center, Scuola Superiore Sant’Anna, 56127 Pisa, Italy; s.brizzolara@santannapisa.it; 5Technology Transfer Centre, Edmund Mach Foundation, Via E. Mach 1, 38010 San Michele all ‘Adige, Italy; duilio.porro@fmach.it; 6Lallemand Italia, Via Rossini, 14/B, 37060 Verona, Italy; fbattista@lallemand.com; 7Interdepartmental Research Centre for Viticulture and Enology (CIRVE), University of Padova, Via XXVIII Aprile 14, Conegliano, 31015 Treviso, Italy

**Keywords:** aroma, dry yeast extracts, grapevine, secondary metabolism, stress responses

## Abstract

Dry yeast extracts (DYE) are applied to vineyards to improve aromatic and secondary metabolic compound content and wine quality; however, systematic information on the underpinning molecular mechanisms is lacking. This work aimed to unravel, through a systematic approach, the metabolic and molecular responses of Sauvignon Blanc berries to DYE treatments. To accomplish this, DYE spraying was performed in a commercial vineyard for two consecutive years. Berries were sampled at several time points after the treatment, and grapes were analyzed for sugars, acidity, free and bound aroma precursors, amino acids, and targeted and untargeted RNA-Seq transcriptional profiles. The results obtained indicated that the DYE treatment did not interfere with the technological ripening parameters of sugars and acidity. Some aroma precursors, including cys-3MH and GSH-3MH, responsible for the typical aromatic nuances of Sauvignon Blanc, were stimulated by the treatment during both vintages. The levels of amino acids and the global RNA-seq transcriptional profiles indicated that DYE spraying upregulated ROS homeostatic and thermotolerance genes, as well as ethylene and jasmonic acid biosynthetic genes, and activated abiotic and biotic stress responses. Overall, the data suggested that the DYE reduced berry oxidative stress through the regulation of specific subsets of metabolic and hormonal pathways.

## 1. Introduction

Grapevine (*Vitis vinifera* L.) is one of the most important horticultural crops in the world, with an annual production of about 90 million tons, of which almost 75% are meant to be transformed into wine [1,2]. Vineyards’ productions are seriously threatened by climate change, especially by the globally average temperature rise, which over the last few years has already manifested a great impact on the wine industry. This scenario is expected to become even more serious in the near future [3,4]. The negative impacts of climate change on grapevines are both quantitative and qualitative. Drought and heat stresses during berry development and ripening result in significant changes in the fruit composition and lower-quality wines [1,5]. Under heat stress, grape berries accumulate sugars and lose their acidity at a faster rate, resulting in wines with excessive alcoholic content and poor acidity. Both primary (sugars, organic acids, and amino acids) and secondary (aroma compounds and phenolics) metabolisms are highly affected by high temperatures during grape berry ripening [6,7]. Previous studies have indicated that elevated temperatures across *veraison* and ripening irreversibly inhibit anthocyanin biosynthesis in the skin of red grapes and consequently result in reduced color in grapes and wines [8]. Additionally, wine aroma is affected by climatic changes [3]. Wine aroma is defined by the complex profile of several volatile molecules belonging to different chemical classes, such as terpenes, lipoxygenase (LOX) pathway products, norisoprenoids, methoxypyrazines, and sulphur (thiolic) compounds [1]. The regulation of the biosynthetic steps of aroma components is under developmental and environmental control, and the diverse aromatic pathways respond differently to drought and heat stress, some of them being enhanced and some being inhibited [3]. Sulphur compounds, referred to as volatile or varietal thiols, which contribute to desirable fruity aromas in Sauvignon Blanc (SB), are particularly sensitive to drought and temperature. Although a moderate water deficit may lead to the accumulation of varietal thiols, extreme drought and high temperatures cause a detrimental decrease in their amount [3]. The concentration of volatile thiols in the must and wine is overall connected with the concentration of their cysteine- or glutathione-bound precursors in the grapes, the biosynthesis of which is closely connected to glutathione (GSH) levels through its reaction with LOX pathway metabolites (e.g., hexenal) [1,9,10,11]. The prevalent sulphur compound (varietal thiol) in the must/wine is 3-mercaptohexan-1-ol (3MH), and its precursors are 3-S-glutathionylhexan-1-ol (GSH-3MH) and 3-S-cysteinylhexan-1-ol (Cys-3MH). GSH-3MH derives from the conjugation of GSH to hexenal through the action of GSH transferase (GST), while Cys-3MH arises from GSH-3MH metabolism by gamma glutamyl transpeptidase (GGT) [12]. GSH levels also display a significant decrease as a consequence of exposure to high temperatures and drought stress, to which GSH is very sensitive due to its use to counteract cellular oxidative stress [3,8]. Among the other aromatic compounds, methoxypyrazines are significantly reduced, while o-aminoacetophenone (AAP), which is a wine-off flavor, is increased when the vines are subjected to heat stress [8]. Monoterpenes are reported to be highly produced in response to high temperatures; however, their excessive amount is detrimental to the end flavor of wine [3]. In this complex context, mitigation strategies are needed to ensure the consistent quality of wines over different vintages. Some tools are available for viticulturists to mitigate the negative effects of climate change and to improve the compositional quality of the berries, including management of the canopy, water supply, nutritional status of the plants, and the supply of elicitors or biostimulants as either foliar sprays or soil applications [5]. However, these measures are, by themselves, not enough to face the rising temperature and lack of water and have limited benefits. According to Van Leeuwen et al. (2016) [4], the most effective way of counteracting climate change is the choice of plant material, but this cannot provide an immediate means to counteract climate change. In addition, a clear classification of the varieties according to drought tolerance still needs significant research to be developed. Irrigation, besides having a high financial and environmental impact, can result in salt accumulation in the soil, creating a new kind of stress and becoming less feasible in a context of limited water availability [4]. The adoption of nebulized water cooling of the canopy to preserve fruit composition and improve the aroma potential of grapes has the same drawbacks as irrigation [13].

Elicitors are one of the most commonly used strategies to increase secondary metabolite content in berries. Previous studies indicated that elicitors stimulate an “immune-like” response in grapes, triggering the production and accumulation of compounds such as polyphenols, phytoalexins, stilbenes, anthocyanins, and tannins [5,9,14]. Inactive dry yeast extracts (DYE) are a group of elicitors that can induce the accumulation of phenols, terpenoids, and other aromatic compounds [5,14]. Thus, DYE treatments are gaining interest as tools that can be exploited in organically managed vineyards to enhance the aroma profile of white grape varieties in the presence of unfavorable climatic conditions. According to Šuklje et al. (2016) [15], DYE applied to SB grapes increased the berry content of antioxidants, in particular GSH, certain amino acids, esters, and the precursors of aromatic thiols [15]. Giacosa et al. (2019) [5] observed an increase in anthocyanin production and berry skin thickness and reported an upregulation of volatile compounds, namely free acetate and ethyl esters, after DYE treatment. Furthermore, the authors proposed DYE as an effective method to counteract some of the consequences of climate change, such as heat stress [5]. While the positive effects exerted by DYE on aroma precursors or secondary metabolites are well documented, very little is known about the biochemical and molecular mechanisms of action of DYE, and a comprehensive transcriptional overview of such effects is lacking. In fact, Pastore et al. (2020) [14] have shown, through a targeted approach, that some key genes (e.g., UFGT) of the anthocyanin pathway are transcriptionally induced by the treatment with DYE in grapevine berries of the red variety Sangiovese. The same authors have hypothesized that this effect may take place as a consequence of an elicitor-like action of DYE, finally leading to the downstream activation of the above-mentioned genes. However, a systematic overview of the transcriptional changes induced by DYE on grape berries is currently missing, and the signaling pathways that may be involved in this process are currently unknown [14].

The purpose of this work was to fill this gap by studying in detail the biochemical and molecular effects of foliar applications of DYE on SB grapes and providing a hint on the molecular signaling pathways that may underpin the improvement in berry aroma and secondary metabolism. Overall, our study aimed to obtain a comprehensive overview, by RNA-Seq, of the transcriptional signatures evoked by the treatment with DYE to provide a systematic molecular frame for their specific mode of action. In particular, we wanted to highlight the transcriptional rewiring of the specific metabolic, hormonal, and oxidative stress responses that are evoked by DYE without impacting the overall ripening process, thus providing a first draft of the DYE elicitor-like molecular action.

## 2. Results

### 2.1. Acidity, Total Sugars, Amino Acids, and GABA

Acidity and total sugars were used as a proxy for the staging of berry ripening progression in treated and control samples. Total titratable acidity and total sugars were measured at each time point during both vintages (Figure 1). The treatment with DYE did not induce significant differences in terms of total sugar content or in the acidity of berry juice, except for acidity at TP2 in 2020, which appeared to be transiently higher in the treated samples. This effect was not confirmed in 2021. In fact, at H, both treated and control samples showed no differences in these parameters.

The complete amino acid profile, as well as GABA (gamma-aminobutyric acid) and other secondary metabolites, have been determined at TP2 and at H in both 2020 and 2021 (the complete dataset is provided in Appendix A). The collective average data is shown in Figure 2.

According to Figure 2A, the greatest differences among samples were observed between control samples at TP2 and at H. The effect of ripening was in fact evidenced by higher contents of cysteine, proline, hydroxyproline, phenylalanine, pipecolic acid, cis- and trans-piceid, and viniferin, and by lower contents of methionine, asparagine, and GABA, independently from the treatment. At H, the DYE application resulted in higher levels of valine, leucine, methionine, isoleucine, and tryptophan and a less marked decrease of arginine, asparagine, and GABA, or, conversely, a less marked increase of cis- and trans-piceid and viniferin. For some of these compounds (valine, tryptophan, and cysteine), this effect was already evident at TP2. At maturity, as a consequence of these amino acid trends, the arginine to proline ratio, a commonly used marker for evaluating berry maturity [17], was higher in treated berries (2.7-fold) in comparison to the control ones (1.4-fold) in the 2020 season.

Figure 2A was confirmed by PCA analysis, which highlighted the separation of samples according to their ripening stage along dimension 1 (accounting for 43.0% of variability) and positioning samples at H on the left side and TP2 samples on the right side of the graph (Figure 2B). Conversely, the second dimension (describing 25.4% of variability) of the PCA enabled the separation of control and treated samples at H, while treated and untreated samples grouped together at TP2, consistent with the previous analysis (Figure 2B). Figure 2C reports the loading plot driving the observed sample clustering, showing that the metabolites contributing the most to the separation of the grouping samples collected at H from the ones sampled at TP2 were the secondary metabolites viniferin, *cis*- and *trans*-piceid and pipecolic acids, and the amino acids proline, hydroxyproline, and phenylalanine. On the contrary, leucine, isoleucine, tryptophan, valine, histidine, and methionine were the amino acids contributing the most to the separation of treated and control samples at H along the y axis.

A targeted T-test analysis carried out on the above-identified amino acids (affected by the treatment in PCA of Figure 2B and significantly different in Figure 2C) pointed out significant differences between treated and control samples (Figure 3).

The accumulation of arginine, lysine, asparagine, tyrosine, and methionine was similarly induced by the treatment with DYE in 2020, displaying significant differences at H. On the other hand, in 2021, the mentioned amino acids showed no significant differences between treated and control samples. At H of the 2021 vintage, proline, phenylalanine, histidine, and isoleucine were significantly reduced by the DYE spraying. Valine and tryptophan had consistent results, with a significant increase in the content of both amino acids at TP2 and H in 2020. The increase in valine content was maintained at TP2 in 2021, while at H, no significant differences could be detected. A significant decrease in tryptophan was recorded at H in 2021. GABA was higher in control samples at TP2 of both years, where in 2020 the decrease exerted by the treatment was significant and in 2021 this trend was maintained even though not significantly. Conversely, at H, GABA had no significant differences in both years.

### 2.2. Aromatic Profile

The aromatic profile has been investigated on the samples at all time points (free volatiles) and at H (bound and thiol aroma precursors) by GC-MS and LC-MS, respectively. In total, 16 free aromas and 20 bound aromas were detected and quantified (Appendix A, respectively). In Figure 4, only the aromas that were significantly affected or that showed a specific trend in response to DYE treatment are reported.

In both years, the treatment with DYE improved the contents of cys-3-Mercaptohexanol (cys-3MH) and GSH-3MH in grapes, even though these differences were not significant (Figure 4A). Glutathione (GSH), an antioxidant and a precursor of GSH-3MH (through a reaction with hexenal) and of Cys-3MH [16], was not significantly affected by the treatment.

Terpenes such as nerol and geranic acid appeared to increase significantly after DYE treatment in 2020 and 2021, respectively. The treatment also affected the LOX pathway by significantly increasing hexanal and decreasing the amount of 2-hexenal (non-significantly), an effect that was especially evident at H. DYE spraying also changed the benzenoid pathway through a slight stimulatory effect on methyl salicylate (Figure 4A) and a significant positive effect on benzeneacetaldehyde (Figure 4B).

### 2.3. Targeted and Untargeted Molecular Responses: Comparison of DYE-Treated versus Untreated Berries

Since DYE treatments result in the enhancement of some aroma components, the transcriptional expression of the main genes involved in berry aroma development [1,12,18] was determined through a targeted approach (RT-qPCR) on berries collected from both years to gain a first characterization of DYE molecular effects. Several genes (in total eight genes) were studied, including those encoding *VvGGT* (Gamma-Glutamyl Transferase) (one gene) and *VvGST* (Glutathione S-transferase) (one gene), selected on the basis of their role in the biosynthesis and metabolization of GSH- and cysteine-bound thiol precursors [12,18]. *VviCCD* (Carotenoid Cleavage Dioxygenase) (one gene) was studied for its role in the biosynthesis of norisoprenoids precursors from apocarotenoids [1,19]. Among these, some displayed a specific expression trend and responsiveness to DYE treatment (shown in Figure 5) (all expression data are reported in Appendix A).

All the considered genes showed an increasing level of expression along with the progression of ripening, reaching a maximum at H. Some of them displayed further increases or decreases of expression in response to DYE, but no interferences with the overall expression dynamics were noticed. In fact, in treated grapes, the transcript levels of the *VvGST3* gene (involved in the biosynthesis of GSH-MH) [18] appeared tendentially higher than those found in control grapes for both years. Conversely, *VvGGT* (responsible for the conversion of GSH-3MH into cys-3MH) [12] showed always more abundant transcripts in control samples, even though more evidently at H in 2021, and *VviCCD4a* was more abundantly transcribed in treated grapes at H in both years.

To have a comprehensive, untargeted overview of the transcriptional profiles induced or repressed by the treatment with DYE, RNA-Seq analysis was performed at TP2 and H in berries collected from the year 2020 (Appendix A). The selection of the samples to analyze was based on the more prominent effects exerted by the DYE treatment on the overall amino acid (Figure 2 and Figure 3) and aroma precursor (Figure 4) profiles in this vintage.

At first, a very stringent analysis was carried out, comparing treated versus control samples at each time point taken independently, to highlight the effects of the treatment without considering the effects of time (ripening). This analysis showed that there were more differentially expressed genes (DEGs) between treated and control samples at H (448 DEGs) rather than at TP2 (118 DEGs), and most DEGs at H were upregulated (301 DEGs) by the treatment, while at TP2 the number of up- or down-regulated genes was similar (54 and 64, respectively) (Figure 6). The majority of DEGs showed differences in the order of 0.6 < log2 < 1 (upregulated) and −1 < log2 < −0.6 (downregulated). Most of the genes identified as DEGs at TP2 were not conserved as DEGs at H (data not shown).

Gene ontology enrichment analysis was carried out to highlight the pathways mainly affected by the DYE treatment by using three different approaches, namely G:Profiler, Mapman, and KEGG. Among the categories identified by G:Profiler within the upregulated genes at H, terms related to protein stability (“unfolded protein binding”, “chaperone binding”, and “protein folding”), transcriptional regulation (e.g., “regulation of transcription” and “DNA-binding transcription factor activity”), stress hormone action (“ethylene-activated signaling pathway”), and abiotic stress (“response to heat” and “response to hypoxia”) were abundantly represented. Among the downregulated genes at H, the categories “ADP and ATP binging”, “protein binding”, “cellular glucan metabolic process”, and “xyloglucan-xylo-glucosyl transferase”, and “defence response” were enriched (Figure 7A,B).

Mapman analyses also identified most of the DEG enrichments at H (Figure 7C). Briefly, “Secondary metabolism of terpenoids: terpene biosynthesis”, “cell division”, “protein homeostasis proteolysis: aspartic-type peptidase activity”, “solute transport: transport channels and primary active transport ABC superfamily”, “response to pathogens and effector triggered immunity”, and “transferases” were downregulated at this time point. Conversely, “Phytohormone action: jasmonic acid”, “RNA biosynthesis: transcriptional regulation AP2/ERF transcription factor superfamily”, “protein homeostasis: protein quality control and ubiquitin-proteasome system”, “protein homeostasis: serine type peptidase activities”, “External response to temperature: cold response and ICE-CBF cold acclimation transcriptional cascade”, were upregulated at H. The enriched categories “Secondary metabolism of terpenoids: terpene biosynthesis” and “protein homeostasis proteolysis: aspartic-type peptidase activity” were downregulated both at TP2 and at H.

KEGG analyses highlighted several biochemical pathways underpinning the previously identified enrichments (data not shown). As far as glutathione and sulphur metabolisms were concerned, an upregulation of glutathione peroxidase (EC 1.111.9), responsible for the conversion of reduced GSH into its oxidized version GSSG, and a downregulation of adenylyl-sulphate reductase (EC 1.8.4.9), leading to cysteine production, were observed. Regarding the regulation of genes potentially involved in plant-pathogen interaction, the transcription of CDPKs (calcium-dependent protein kinase), CaM/CML (calcium sensors involved in the regulation of plant development and stress responses), HSP90 (heat shock protein 90), and RBOHs (respiratory burst oxidase, responsible for ROS generation) was upregulated. Within the same pathway, the upstream components CNGSs (cyclic nucleotide gated channels) and RPM1 (NBS-LRR protein) were downregulated. Furthermore, a gene encoding WRKY229 and involved in the MAPK signaling pathway, linked to H_2_O_2_ production and leading to programmed cell death, was upregulated.

The treatment clearly impacted plant hormonal signaling by downregulating the transcription of two genes encoding a protein homologous to AUX1 (auxin influx transporter) and CRE1 (cytokinin receptor 1), respectively [20,21], while a gene encoding a putative ABA receptor (PYR/PYL) was upregulated. Carotenoid biosynthesis and metabolism, connected with ABA biosynthesis, were affected through the downregulation of a gene encoding zeaxanthin epoxidase and the upregulation of a gene encoding an epoxicarotenoid dioxygenase, the latter enzyme being the limiting step responsible for producing xanthoxin and, consequently, abscisic acid (ABA). The transcription of a gene encoding jasmonate o-methyltransferase was upregulated, meaning that the conversion of jasmonate into methyl-jasmonate may be favored. Several genes encoding ABC transporters were differentially regulated: ABCB (participating in the mediation of polar auxin transport and multidrug resistance), ABCC (associated with detoxification), and ABCG (PDR5) subfamily members were down-regulated, while an ABCG subfamily member (responsible for secondary metabolite transport) was up-regulated (Appendix A).

Besides the previously mentioned pathways, KEGG analysis also highlighted transcriptional changes related to amino acids. Genes encoding acetolactate synthase and L-proline dehydrogenase, which are involved in the pathway of biosynthesis of L-valine, L-leucine, and L-isoleucine and in the conversion of proline into arginine, respectively, were downregulated. Moreover, the biosynthetic pathway of mugineic acid (an iron chelating compound) from methionine was altered after the DYE application through the upregulation of nicotianamide synthase. The gene encoding hydroxypyruvate reductase, belonging to the serine biosynthetic pathway and responsible for the transformation of hydroxypyruvate into glycerate, was upregulated. Finally, in the nucleotide metabolism, the transformation of uridine (a by-product of L-glutamine) into uracil was upregulated (Appendix A).

### 2.4. Untargeted Analysis of Molecular Responses to DYE Application: Combined Effects of Time and Treatment

The effect of the treatment on the overall time-course expression of genes during the ripening of the berry was analyzed by comparing the dynamics of the transcriptional profiles from TP2 to H in treated and control samples using the likelihood ratio test (LRT) in DESeq2. LRT allowed the identification of all genes that showed a change of expression across the different timepoints as a function of the effect of treatment with DYE on their expression trends during berry ripening.

After statistical analysis (Figure 8), it was possible to cluster the genes into four groups according to their transcriptional expression behavior. In the three groups 1, 2, and 4, genes that, in untreated berries, displayed an intrinsically slightly decreasing (groups 1 and 4) or clearly decreasing (group 2) expression trend along with the ripening of the fruit were included. Conversely, group 3 included genes that displayed an upregulation from TP2 towards ripening (H) (Figure 8A,B). The treatment with DYE interfered with these naturally occurring gene expression trends by smoothing or even reversing them. Genes belonging to groups 1 and 2 were in fact up-regulated by the treatment at both time points in comparison to untreated berries, and their decreasing expression trend towards H was prevented for group 2 while it was reverted for group 1, the latter displaying a progressively increasing expression trend at H in the DYE-treated berries (Figure 8A,B). For groups 3 and 4, which included genes that displayed an up or downregulation from TP2 to H, respectively, the treatment resulted in an overall downregulation of the transcriptional expression level of the genes in comparison to their untreated control counterparts. This effect did not evidence a substantial interference of the treatment, with the expression trends over time remaining a progressive downregulation towards ripening for genes belonging to group 4 and an upregulation for genes belonging to group 3. An enrichment analysis was performed for each group. Group 1 (including 225 DEGs) showed significant enrichments for the terms “response to chitin” and “respiratory burst involved in defence response” (Figure 8C). Group 2, probably due to a lower number of genes (41 DEGs), did not highlight any enrichment results. Group 3 (64 DEGs) evidenced the significantly enriched terms “terpene synthase activity” and “terpenoid biosynthetic process”, “lyase activity”, and “magnesium ion binding”. Finally, group 4 was enriched for the term “vacuolar membrane”.

An in-depth analysis of the putative function of the DEGs belonging to the four groups pointed out a selective transcriptional modulation of several genes playing a role in abiotic and biotic stress responses and in specific hormonal metabolic and transduction processes in response to DYE treatments (listed in Table 1). In fact, two genes (*Vitvi08g02053*, *Vitvi08g02051*, group 1) were identified that encode proteins with homology to E3 Ubiquitin Ligases (U-box 23), are putatively involved in the regulation of responses to drought stress and to PAMP-triggered immunity [22,23], and belong to the “response to chitin” and “respiratory burst involved in defence response” enriched categories of group 1. A third U-box protein encoding gene (U-box 17) (*Vitvi12g00139*, group 4), possibly involved in defenses against pathogens [24], appeared to be downregulated. Group 1 also included two WRKY transcription factors, namely WRKY30 (*Vitvi15g01003*) and WRKY33 (*Vitvi08g00793*), identified in Arabidopsis as positive regulators of abiotic responses and resistance to Botrytis, respectively [25,26]. Remarkably and consistently, with the activation of biotic and abiotic stress responses, several genes involved in the modulation of ROS homeostasis were upregulated. These include one gene encoding a glutathione peroxidase (*Vitvi02g00332*, group 1), counteracting oxidative damage [27], one encoding a redox-responsive AP2/ERF transcription factor (ERF109) (*Vitvi03g00500*, group 1) required for activation of ROS quenching genes and proteins, and a respiratory burst oxidase (RBOHB) (*Vitvi14g00183*, group 2) encoding a gene required for the H_2_O_2_-dependent induction of thermotolerance through the stimulation of heat shock proteins [28,29]. Accordingly, several abiotic stress- and ROS-inducible genes encoding heat shock proteins, with putative roles in thermotolerance, were upregulated (e.g., HSP 17.4 (Vitvi13g00491, group 1), HSP 70.1 (*Vitvi06g00443*, group 2; *Vitvi08g02189*, group 1) and HSP 90.1 (*Vitvi02g00025* and *Vitvi16g01103*, group 2)) together with a heat shock transcription factor A2 (*Vitvi04g00092*, group 1), putatively involved in the activation of heat stress memory genes [30], a heat shock transcription factor A6B (*Vitvi07g00078*, group 1), involved in ABA inducible thermotolerance [31] and a histone chaperone protein (anti silencing function ASF1 (*Vitvi01g00372*, group 1)) responsible for the activation of heat stress responses through histone modifications [32] (Table 1).

Interestingly, the DEGs upregulated by DYE treatment (included in groups 1 and 2) also comprised several genes encoding key enzymes of the main biosynthetic steps of jasmonic acid (JA) and ethylene, which act synergistically in response to abiotic and biotic stress, and, to some extent, of ABA and IAA. As far as JA biosynthesis is concerned, two genes encoding lipoxygenase (LOX) (*Vitvi13g01780* and *Vitvi09g00085*, group 1), involved in the early biosynthetic steps of the hormone through metabolism of free fatty acids [43,44], were up-regulated in concert with genes encoding two GDSL-like lipases (*Vitvi09g00038* and *Vitvi10g00669*, group 1) [48], a DAD1-like lipase (Vitvi07g00039, group 1) [49], and a phospholipase (*Vitvi15g00298*, group 1) [50]. These may likely control the upstream degradation of lipid molecules to provide free fatty acid substrates for subsequent LOX action. JA-responsive genes were also upregulated by the treatment, including genes encoding a jasmonic acid carboxyl methyltransferase (*Vitvi04g02169*, group 1), catalyzing the methylation of JA [46,47], and one gene encoding sulfotransferase 2A (*Vitvi13g00864*, group 1), involved in the sulfonation of hydroxy-JA [51]. Among the downregulated genes (groups 3 and 4) putatively involved in JA responses, a second gene encoding a sulfotransferase 2A (*Vitvi13g01379*, group 3) and a JA-responsive gene encoding proline dehydrogenase (PDH2) (*Vitvi14g01283*, group 4) [52], together with genes encoding a MAPK kinase (*Vitvi12g02545*, group 4), which negatively regulate wound responses [53], and a leucine rich repeat receptor kinase (*Vitvi11g01479*, group 4), which represses ethylene and JA signaling pathways [54], were identified. Consistently, a synergic co-regulation of ethylene-related genes could be evidenced, with the upregulation of a gene encoding the ethylene-forming enzyme ACO4 (*Vitvi10g02409*, group 1) [55] as well as various ERF transcription factors encoding genes, showing homology to ERF4 (*Vitvi19g01784*, group 1) (ethylene and JA inducible) [59], ERF5 (*Vitvi16g01438*, group 1) (a positive regulator of JA/ethylene responsive genes to enhance resistance to Botrytis) [56,57], ERF9 (*Vitvi12g00274*, group 1) (regulating defense mechanisms against necrotic fungi through the ethylene/JA signaling pathway), ERF105 (*Vitvi16g01444*, *Vitvi17g00787*, *Vitvi16g01434*, *Vitvi16g01432*, *Vitvi18g00295* and *Vitvi16g01423*, group 1) (regulator of cold responses) [66], and ERF71 (*Vitvi07g00357*, group 1) (hypoxia responsive) [62]. In addition, a phosphatase (AP2C1) (*Vitvi06g00667*, group 1) negatively regulating the ethylene signaling kinases MPK4 and MPK6 [60] and two hypoxia-inducible WRKY-encoding genes (WRKY22 (*Vitvi15g01090*, group 1) and WRKY33 (*Vitvi08g00793*, group 1)) [63,64,65] were upregulated. The DYE treatment also displayed effects on ABA biosynthetic and signaling genes, triggering the upregulation of ABA responsive genes such as a LEA encoding gene (*Vitvi15g01084*, group 1) [75], a cystatin gene (*Vitvi18g03100*, group 1) [79], a DREB subfamily A-1 of the ERF/AP2 transcription factor gene (CBF3) (*Vitvi16g00941*, group 1), and a sulphate transporter gene (*Vitvi05g00548*, group 1) required for the activity of the ABA biosynthetic enzyme aldehyde oxidase [78], a receptor-like kinase gene (LRK10) (*Vitvi01g00353*, group 2) [80], and an SNF1-related kinase (*Vitvi17g00116*, group 1) [67] positively regulate ABA responses. Further positive regulators of ABA signaling that were upregulated by DYE treatment included a gene encoding a WRKY6 (*Vitvi10g00063*, group 1) [73] and a WRKY44 transcription factor (*Vitvi08g00793*, group 1) [69], and an NCED4 carotenoid cleavage dioxygenase (*Vitvi02g01288*, group 2), the latter probably not involved in the direct generation of ABA [77], as well as a cytokinin responsive factor that was claimed to play a role in nitrogen uptake [93]. On the other hand, genes encoding a putative ABA biosynthetic enzyme NCED6 (*Vitvi05g00963*, group 3) [83], the ABA transporter ABCG25 (*Vitvi18g01703*, group 3) required for fine-tuning ABA export from cells, the ABA importer ABCG40 (*Vitvi04g00423*, group 3) [82], and the ABA antagonistic and gibberellin inducible gene GASA6 (*Vitvi17g00601*, group 3) [84], as well as a gene encoding a cytokinin receptor histidine kinase (*Vitvi12g00685*, group 4), a negative regulator of ABA sensitivity, were downregulated by DYE treatment. Finally, DYE treatment resulted in the overall downregulation, in comparison to untreated berries, of a gene encoding an AUX1 auxin influx carrier (*Vitvi13g00019*, group 4) [90] and one gene encoding a PIN1 auxin efflux carrier (*Vitvi17g00210*, group 3) [94], and in the upregulation of an Aux/IAA encoding gene (AUX/IAA7) (*Vitvi11g00394*, group 2) [86] and a Ca^2+^-dependent transducer of auxin responses (*Vitvi12g00594*, group 1) [87].

The most relevant functional results of the untargeted molecular analysis of genes listed in Table 1 are schematically represented in Figure 9. Overall, DYE treatment actively changed the expression (down and/or upregulated) of genes participating in sulphur and glutathione metabolism, plant-pathogen interaction, ROS signaling, hormone biosynthesis and signal transduction, and hormone transporters.

## 3. Discussion

Aromatic development in white grapes is the result of the activity of a wide range of metabolic pathways balanced by the interplay between developmental and environmental cues [1]. Several parameters play crucial roles in determining the final grape composition in terms of sugars, acidity, and the accumulation of secondary metabolites and, consequently, in wine quality. These include genetic factors, vineyard management, soil physico-chemical composition, and climatic factors. Secondary metabolites have a great impact on wine aroma, which in turn is recognized as one of the main factors in wine’s quality [1,5]. The evident impacts of climate change on grape berry aroma and composition are complex and vary depending on the specific components of the aromatic profiles [6,7]. Maintaining suitable compositional characteristics for later transformation into wines with well-defined and recognizable sensory attributes is an urgent need for wine makers and markets, requiring the implementation of effective mitigation strategies. The adoption of naturally derived molecules to enhance the overall aroma complexity of grapes or to keep it at acceptable levels even in the presence of the negative effects of adverse climatic conditions represents an important tool for viticulturists [6,7,8]. The use of DYE applications on grapes has been recently proven to be an effective strategy to ensure the maintenance of secondary metabolite content. For example, in Merlot grapes, Tomasi et al. (2021) [95] have shown that DYE treatments increased overall flavonoids and anthocyanin contents and their extractability [95]. Consistently, Pastore et al. (2020) [14] have reported that DYE treatments stimulated the biosynthesis of anthocyanins in the berries of Sangiovese potted plants, leading the authors to conclude that DYE may be an effective mitigation strategy to counteract the detrimental effects of high temperatures on anthocyanin biosynthesis [14]. Similarly, additional studies performed on white grapes demonstrated that the DYE application on grapes may also be effective in improving or better maintaining the aromatic precursors in Glera [96]. In Sauvignon Blanc (SB), these effects included a stimulation of GSH and of the thiol aroma precursors of 3-mercaptohexanol (3MH) typical of Sauvignon Blanc grapes, such as cysteine- and glutathione-bound cys-3MH and GSH-3MH, respectively [10]. Even though Pastore et al. (2020) [14] have shown that in red grapes, DYE exerts its action through the upregulation of the key genes involved in the anthocyanin biosynthetic pathway (e.g., PAL (phenylalanine ammonia lyase), CHS and CHI (chalcone synthase and isomerase), and F3H (flavanone 3-hydroxylase)), the mechanisms of action of DYE are still only partially clarified [14].

In this manuscript, we have aimed at reaching a more comprehensive view of the physiological and molecular mode of action of DYE in white grapes. To this end, we have treated SB’s vineyards for two consecutive years (2020 and 2021) with specific inactive dry yeast extracts. Following the application, we have characterized as comprehensively as possible the evolution of ripening, the contents of the main berry metabolites, and the transcriptional profiles. Our data confirmed previous findings suggesting that DYE did not interfere with the overall technological ripening in that, at H, acidity and total sugars showed no significant differences between treated and control samples [15]. Therefore, DYE treatments did not significantly interfere with the overall ripening process of SB grapes in the years considered. Thus, DYE action seems to depend on more specific effects on selected metabolic pathways.

In fact, cys-3MH and GSH-3MH (the precursors strongly influencing the aromatic profile of SB’s wine), terpenes (nerol and geranic acid), and benzenoids (methyl salicylate) increased in the berries of DYE-treated SB grapevines, in agreement with the findings of Šuklje et al. (2016) [15]. Conversely, GSH content slightly decreased after DYE treatment, even though no significant differences were observed. In plants, glutathione contributes to sulfur metabolism, redox control, and detoxification, and in SB berries, it contributes to the biosynthesis of the above-mentioned 3MH precursors by glutathionylation of trans-2-hexenal, a product of the LOX pathway [18]. Gamma glutamyl transpeptidase (GTT) is responsible for the conversion of GSH-MH into Cys-MH [12]. Indeed, the higher and lower expression, respectively, of the *VvGST3* and *VvGGT* encoding genes (Figure 5), and the above-described levels of the respective metabolites, may support the hypothesis of a stimulation of GSH conjugation with trans-2-hexenal in response to DYE. LOX pathway-derived products are often described as having an herbaceous, leafy “green” aroma in wine [97], thus the stimulation of their conjugation with GSH may also result in a reduced herbaceous character. Thus, our findings suggest for the first time that DYE treatment may increase the activity of the LOX pathway and induce a faster turnover of 2-hexenal and GSH through the regulation of GST and GGT. 

The analysis of the complete profiles of amino acids showed that the treatment with DYE overall did not strongly interfere with the evolution trends of the amino acid contents during ripening, which showed a progressive increase towards later ripening stages (from TP2 to H) for valine, methionine, proline, and tryptophan and a progressive decrease for arginine, asparagine, and GABA (Figure 3). When analyzing the single compounds, arginine and proline remained the most representative amino acids, accounting for 67% of total amino acids both in control and treated berries at ripening, although with different percentages in the two theses (Figure 3B). Nevertheless, DYE induced a delay in arginine decrease as well as in proline increase, resulting in a higher arginine-to-proline ratio at H. A higher arginine-to-proline ratio means a balanced proportion of the yeast-assimilable amino acids that are required to carry on a satisfying fermentation, particularly when a yeast strain with high nitrogen requirements is used [98]. Similarly, valine was more accumulated at most time points after DYE spraying during both years (2020 and 2021). According to Stribny et al. (2015) [99], valine is degraded into isobutyl alcohol, an aliphatic alcohol that is fermented by yeast, giving a desired fruity and floral aroma to wines [99]. The increase of this amino acid in grapes after DYE treatment may result in an improvement in wine aroma and is consistent with the effects suggested by Šuklje et al. (2016) [15], reporting specifically on the effects exerted by DYE on berry amino acidic composition and its potential consequences on yeast fermentation [15]. Similarly, benzeneacetaldahyde, which is synthesized from phenylalanine and appreciated in the sensory profile of wines, providing sweet flowery notes [100,101], significantly increased in response to DYE. The downregulation of phenylalanine in one vintage may be explained because of its transformation into benzeneacetaldahyde. Proline has been suggested to play an adaptive role in plant stress tolerance, acting as a ROS scavenger, stabilizing proteins’ structures, and signaling stress [102]. The observed reduction in proline levels induced by DYE treatment may be an indication of reduced berry stress. Similarly, GABA is a molecule associated with responses to abiotic and biotic stress factors [35,103]. The decrease in GABA may further confirm the above findings and suggest that DYE could improve the overall health of the vine.

These hypotheses were also supported by the untargeted RNA-Seq molecular analyses carried out on samples of the 2020 vintage, through: (1) the comparison of treated versus untreated berries at each time-point (TP2 and H); and (2) the evaluation of the DYE effects on the dynamics of ripening-associated genes. The latter analysis allowed the identification of four different groups of DEGs clustered according to their ripening specific expression profiles and significantly affected by the treatment in terms of either an increased or decreased expression level (Figure 8). For one group of genes (Group 1, Figure 8), the DYE treatment, besides resulting in a significant stimulation of transcription at both TP2 and H, reverted the trend of expression along ripening from a downregulation into an upregulation. Indeed, one of the main findings of this work was the identification of several genes involved in abiotic and biotic stress responses, highlighted as differentially expressed in later analyses. The identification of the term “response to chitin” in the upregulated DEG enrichment strategy suggests the elicitor-like mode of action of DYE, as well as the enrichment terms “response to heat”, “response to hypoxia”, and “response to pathogens and effector triggered immunity” clearly indicated the stimulation by DYE of responses that recalled those typical of abiotic and biotic stresses (Figure 7 and Figure 8). Since chitin is a ubiquitous cell wall structural component of fungi and is contained in DYEs, plants have chitin-specific receptors that activate defense mechanisms, relying on phenolics, terpenes, and ROS production [104,105]. Our RNA-seq data support the hypothesis that DYE affected ROS homeostasis in the berries, as testified by the upregulation of a gene encoding glutathione peroxidase (*Vitvi02g00332*, group 1), responsible for the transformation of glutathione (GSH) into its oxidized form glutathione disulfide (GSSG) to scavenge H_2_O_2_ in excess. The GSH:GSSG ratio is in fact an important bioindicator of cellular oxidative stress [27,106]. The DYE treatment also stimulated sulfate metabolism by activating the transcription of the gene encoding APS (adenosine phosphosulfate), which is involved in the production of cysteine, which in turn contributes to GSH levels. This is in accordance with the tendentially higher (even though not significantly higher) contents of cysteine observed in treated samples in the 2020 vintage (supporting data, Appendix A). Furthermore, as reported above, the concomitant upregulation of several genes involved in the modulation of ROS homeostasis, such as that encoding the Redox Responsive AP2/ERF transcription factor ERF109 (*Vitvi03g00500*, group 1) (activator of ROS quenching genes) and a respiratory burst oxidase (RBOHB) (*Vitvi14g00183*, group 2) generating H_2_O_2_, led us to conclude that DYE is able to trigger an augmented ROS biosynthesis/scavenging loop and, consequently, an improved abiotic stress tolerance of berries (Table 1). This protective effect could depend on the downstream upregulation of a subset of ROS-inducible genes encoding proteins involved in controlling thermotolerance, such as heat shock proteins that were previously shown to be ROS-inducible (e.g., HSP 17.4, 70.1, and 90.1) (*Vitvi13g00491*, *Vitvi08g02189*, *Vitvi09g00045*, group 1; *Vitvi06g00443*, *Vitvi02g00025*, *Vitvi16g01103*, group 2) [36,38,40], NO-inducible (heat shock transcription factor A2) (*Vitvi01g01846*, group 1) [30], ABA-inducible (A6B) (*Vitvi07g00078*, group 1), or heat stress-inducible (anti silencing function ASF1) (*Vitvi01g00372*, group 1) [42]. These genes share a common transcriptional responsiveness in the presence of ROS, so this may be considered a downstream consequence of the above-mentioned increased ROS homeostatic levels evoked by the DYE treatment. In addition, the same genes are involved in the long-term regulation of heat stress responses and thermotolerance through the activation of heat stress memory genes and histone modifications [32].

Considering these data together, our findings provide for the first time significant molecular evidence for the previously hypothesized (Pastore et al., 2020) [14] elicitor-like mode of action of DYE on grapevine berries. At the same time, our data also highlight the putative signaling pathways of DYE responses by identifying several transcription factors, some of which appear to be ROS-responsive, that may be responsible for the downstream regulation of specific metabolic genes.

In addition, our data would support the hypothesis that the berries of plants subjected to DYE treatment display increased resilience to abiotic stress. This effect may be accomplished through a finer regulation of ROS levels and a reduced oxidative stress that may be testified by a diminished proline content, assumed as a proxy for oxidative stress [102], and by the upregulation of the *VviCCD4a* encoding gene, most likely not involved in ABA biosynthesis but rather in the generation of signaling molecules [77]. In fact, preventing the oxidation of aroma precursors in grapevine berry fruits, especially when they are exposed to heat or drought, either alone or in combination, has been claimed by several authors to be essential to favor better later maintenance in the wine [4,5,14]. 

Moreover, we found that the treatment with DYE specifically induced a subset of hormonal responses, a novel, unpredicted aspect that has not been evidenced by previous studies dealing with DYE. In particular, the upregulation of several genes encoding key enzymes of the main biosynthetic steps of jasmonic acid (JA) and ethylene strengthens the hypothesis of a selective “priming” action of DYE. This priming action depends on the increase of these two stress hormones, which notoriously act in synergism in the regulation of stress responses. It can be speculated that these hormones may be possibly involved in reinforcing those responses falling within the functional categories of “response to chitin” and “response to pathogen”, while leaving unaffected the main ripening processes. Regarding JA biosynthesis, the upregulation of three types of lipases (*Vitvi09g00038*, *Vitvi10g00669*, *and Vitvi15g00298*, group 1) [48,49,50] and of two LOX genes (*Vitvi13g01780*, *Vitvi09g00085*, group 1) [44,45] supports the view of stimulation of several key biosynthetic steps of the hormone. The upregulation of JA-responsive genes like hydroxy-JA sulfotransferase 2A (*Vitvi13g00864*, group 1) [51] may be a feedback control response induced by the increased JA levels. The activation of JA responses was also confirmed by the downregulation of negative regulators of JA signaling (Table 1). The co-regulation of the ethylene biosynthetic gene ACO4 (*Vitvi10g02409*, group 1) [55] as well as various ethylene and JA inducible ERF transcription factors encoding genes (ERF4 (*Vitvi19g01784*, group 1) [59], ERF5 (*Vitvi16g01438*, group 1) [56,57], ERF9 (*Vitvi12g00274*, group 1), and ERF105 (*Vitvi16g01444*, *Vitvi17g00787*, *Vitvi16g01434*, *Vitvi16g01432*, *Vitvi18g00295* and *Vitvi16g01423*, group 1) [66]) further suggests the simultaneous enhanced action of the two hormones induced by DYE treatment. It is interesting to note that some of these genes, together with WRKY encoding genes (WRKY22 (*Vitvi15g01090*, group 1) and WRKY33 (*Vitvi08g00793*, group 1)) [63,64,65], were reported in Arabidopsis to be putatively associated with hypoxia responses and with JA- and ethylene-inducible resistance to necrotrophic pathogens (*Botrytis*), further supporting the hypothesis of a “priming” effect exerted by DYE.

ABA responses appeared to also be affected, although the final readout of such effects seems less univocally interpretable since both positive and negative regulators were induced by the treatment. The upregulation of ABA responsive genes (Table 1), as well as the increased transcription of a sulphate transporter gene (*Vitvi05g00548*, group 1) required for ABA biosynthesis [78] and of an SNF1-related kinase gene (*Vitvi17g00116*, group 1) [68], and the downregulation of a gene encoding a cytokinin receptor histidine kinase (*Vitvi12g00685*, group 4) negatively regulating ABA sensitivity, would support an overall positive regulation of ABA responses by DYE supply. However, this hypothesis may be contrasted by the downregulation of a gene encoding the rate-limiting step of ABA biosynthesis, NCED6 (*Vitvi05g00963*, group 3) [83]. It is interesting to note that the DYE treatment also probably interfered with the transport of two hormones: of ABA, by down-regulating the expression of genes with similarities with two ABA transporters (the ABA export carrier ABCG25 (*Vitvi18g01703*, group 3) and the ABA importer ABCG40 (*Vitvi04g00423*, group 3) [82]), and of auxin, by down-regulating genes encoding proteins with similarity to the auxin influx carrier AUX1 (*Vitvi13g00019*, group 4) [90] and the efflux carrier PIN1 carrier (*Vitvi17g00210*, group 3) [94]. An interesting effect of DYE is also the reversion of declining expression of a cytokinin-responsive factor (CRF4) (*Vitvi16g01201*) from TP2 to H (Table 1). This gene has been claimed to be a master regulator of nitrogen uptake by controlling more than half of the genes associated with nitrogen uptake and assimilation [107]. By considering that nitrogen regulation can effectively promote the improvement of berry components and the formation of flavor compounds in wine grapes [108], the fact that DYE treatment can modulate the expression of CRF4 is a topic that should be further investigated.

Overall, with these data, we provide the first comprehensive picture of the transcriptional signaling pathways that are modulated by the treatment with DYE in grapevine berries. We identified for the first time several DYE-responsive transcription factors that may be responsible for the activation of the downstream metabolic responses of berries selectively modulated by DYE without interfering with the main ripening parameters, sugars, and acidity. In addition, our data also provide novel and significant insights supporting the role of DYE in the modulation of ROS stress and, unexpectedly, of specific subsets of the signaling components of the JA and ethylene hormonal pathways, activated to modulate stress adaptation through molecular priming. Finally, both the molecular and metabolic evidence provided in this work point out a specific effect of DYE on the LOX pathway that may be coupled to a faster turnover of GSH, explaining the DYE-dependent increase of GSH- and Cys-bound 3MH precursors. 

## 4. Materials and Methods

### 4.1. Experimental Design and Plant Material

A commercial vineyard (La Madunina), located in northern-east Italy (Friuli-Venezia Giulia region, Sequals, PN), of Sauvignon Blanc (*Vitis vinifera* L.) was chosen to test a commercial DYE product (“LalVigne™ AROMA”, Lallemand Inc., Montreal, QC, Canada) in 2020 and 2021. The choice of the experimental sites was based on the high degree of homogeneity of the vineyards, on the presence of long rows (200 m) allowing sampling from several independent plants arranged in randomized blocks without impacting the fruit load, and on the fully mechanized conduction of the vineyard, further ensuring homogeneity. The meteorological data (Appendix A) in Sequals was evaluated and determined to show no significant differences in terms of temperature or annual precipitation between the years 2020 and 2021 (1941 and 1971 mm, respectively). During the summer months (July to September), there was a slight change in precipitation during the two vintages (in 2020, approximately 500 mm, while in 2021, around 370 mm).

Six rows were treated, and an additional six rows were kept without treatment and used as a control. The two groups of six rows were separated by an additional two rows to avoid the drift effect of the product. All the treatments were carried out in the early hours of the morning through a double tunnel recovery atomizer adopting the recommended dosage (10g/L; 3 Kg/Ha) [109].

The treatments with DYE were performed as indicated in Figure 10. In 2020, the first treatment was carried out when 5% of grapes reached veraison (⁰Brix ≥ 7) [109]; the second treatment was carried out 11 days later, and the sampling was performed 24 h after each treatment. Finally, a further sampling was carried out on the day of harvest (H). In 2021, as in 2020, the first treatment was executed at the beginning of veraison. A second DYE treatment (TP2) was performed 10 days after the first treatment, and sampling occurred 48 h after each treatment and at H. In this latter vintage, a third sampling was performed between the second treatment and H.

Three biological replicates were sampled for each experimental condition (DYE-treated and control) at each time point (TP). Each replicate was composed of a mix of berries sampled from three independent blocks, each consisting of two rows: the first and second row, the third and fourth row, and the fifth and sixth rows originated from replicates one, two, and three, respectively. From each block, five bunches were collected from each row, so that each replicate was finally obtained from a mixture of berries from ten bunches. Bunches were always collected from the same side of the row, guaranteeing that they were grown under identical sun exposure conditions. The samples were frozen with liquid nitrogen immediately after collection in the field, transported on dry ice, and then stored at −80 °C for further analysis.

### 4.2. Acidity and Total Sugars

Total sugars were measured for each experimental condition individually on 60 berries belonging to each replica/treatment at each sampling date. The measurements were performed in the field using a digital refractometer (Pocket Refractometer Pal-1, Atago (Tokyo, Japan)).

Acidity was determined by titration. For each replica, 20 randomly collected berries were manually pressed, and their juice was collected. Titration was performed in triplicate for each replica, and the total acidity was expressed in grams of tartaric acid per liter as described by Thermo Fisher (Waltham, MA, USA) [110].

### 4.3. Aroma Precursors

#### 4.3.1. Free Precursors

Free volatile aroma precursors have been analyzed at Scuola Superiore Sant’Anna (Pisa, Italy) employing gas chromatography coupled with mass spectrometry (GC-MS) equipment. The solid-phase micro extraction (SPME) technique has been used to analyze the head space of grape samples, applying an untargeted analytical approach to achieve grape berry volatile profiling. Grapes were homogenized with 1 M NaCl buffer solution (1:1 ratio in weight) by using an UltraTurrax (Mod. T25, IKA) and thawed at 15 °C for 15 min. 10 g of sample were transferred to a 20 mL glass crimp vial for headspace analysis and sealed. The vials were incubated under agitation for 45 min at 40 °C, and volatiles were sampled using an SPME fiber (50/30 µm, DVB/CAR/PDMS, 1 cm long; Supelco, Bellefonte, PA, USA). Sample analysis, as well as compound identification and quantification, have been performed according to Modesti et al. (2023) [111]. A Clarus 680 gas chromatograph equipped with a split/splitless injector (PerkinElmer^®^, Waltham, MA, USA) was used for the analysis. Volatiles were separated on a fused silica capillary column (DBWax, 60 m, 0.32 mm ID, 0.25 µm film thickness; Restek, Bellefonte, PA, USA), using helium as carrier gas with a flow rate of 1 mL/min. Compounds were identified using a mass spectrometer (Clarus 500 mass spectrometer, PerkinElmer^®^, Waltham, MA, USA) coupled to the GC. 

#### 4.3.2. Bound Precursors and Thiol Precursors

Bound aromas and thiol precursors were determined at Fondazione Edmund Mach (FEM, San Michele all’Adige, Italy). Bound aromas were extracted and quantified by gas chromatography, according to Paolini et al. (2018) [112]. The extraction was performed by adsorption on an SPE cartridge (ENV+, 1 g) previously conditioned with 20 mL of methanol and 20 mL of Milli-Q water. The sample was loaded onto the cartridge, which was washed with 20 mL of Milli-Q water and eluted with 30 mL of methanol. This solution was evaporated and then dissolved in 4 mL of citrate buffer at pH 5. 200 μl of a glycosidic enzyme with strong glycosidase activity (AR 2000 at 70 mg/mL in water) were added, and the solution was kept in a 40 °C water bath overnight. After the addition of 1-heptanol (0.1 mL) as an internal standard, the free volatiles released were extracted using an SPE cartridge (ENV+, 200 mg) previously conditioned with 10 mL of methanol and 10 mL of Milli-Q water and eluted with 4 mL of dichloromethane. The organic phase was dried. Volatile organic compound analysis was performed using an Agilent Intuvo 9000 fast GC system coupled with an Agilent 7000 Series Triple Quadrupole MS equipped with an electron ionization source operating at 70 eV. The filament current was 50 μA. Separation was obtained by injecting 2 μL in split mode (1:5) into a 20-m DB-Wax Ultra Inert (0.18 mm ID × 0.18 μm film thickness) with He as carrier gas (at a flow of 0.8 mL/min). The oven temperature was programmed starting at 40 °C for 2 min, raised to 55 °C by 10 °C/min, then raised to 165 °C by 20 °C/min, and finally raised to 240 °C by 40 °C/min and held at this temperature for 5 min. The mass spectra were acquired in multiple reaction monitoring modes, setting the instrument within the dynamic system. Nitrogen was used as the collision gas, with a flow of 1.5 mL/min. The transfer line and source temperature were set at 250 °C and 230 °C, respectively. 

Thiol precursors were estimated using liquid chromatography, according to Tonidandel et al. (2021) [113]. The QuEChERS method was used to extract the sample with an extraction buffer composed of acetonitrile. After incubation at 8 °C for 2 h, a 1ml aliquot of the supernatant sample extract was transferred to an autosampler vial (volume 2 mL), and 75 μL of Ebselen (ACN solution at 500 mg/L) was added. The vial was then stirred for 10 min. The derivatized sample was filtered (0.22 μm PTFE) and then transferred into an auto-sampler vial before the injection. A Waters Acquity UPLC (Waters Corporation, Milford, MA), coupled to a Xevo TQ MS mass spectrometer equipped with an electrospray ion source (Waters), was used. The chromatographic module consisted of an Acquity UPLC BEH C18 column (1.7 μm, 2.1 mm ×100 mm) working at 40 °C. 0.1% (*v*/*v*), aqueous formic acid (solvent A), and 0.1% (*v*/*v*) formic acid in MeOH (solvent B) were used at a flow rate of 0.45 mL/min. The gradient conditions of the LC mobile phase were as follows, based on times (t): t = 0–0.25 min, hold 80% A, 20% B; t = 0.25–7 min, ramp linearly to 100% B; t = 7–8.5 min, hold 100% B. The sample (2 μL) was injected using the partial loop needle overfill’ function. The mass spectrometer was used in positive electrospray ionization mode (ESI+), and the source conditions were set as follows: capillary voltage 0.6 kV; source temperature 150 °C; cone gas flow (nitrogen, 20 L/h); desolvation gas flow (nitrogen, 1000 L/h); desolvation temperature 500 °C. Argon, used as collision gas, was set at 0.20 mL/min.

### 4.4. Relative Quantification of Amino Acids, GABA, and Secondary Metabolites

Metabolites were analyzed by UHPLC using an adapted methodology from Schilling et al. (2022) [114]. Metabolites from powdered grape samples (300 mg fresh weight, obtained through the grinding to a fine powder of grape berry skin and pulp with liquid nitrogen) were extracted using 1200 µL of methanol containing 5 µg/mL of chloramphenicol as an internal standard. The extract was then incubated in an ultrasound bath for 10 min before centrifugation at 13,000 g at 10 °C for 10 min. Supernatants were analyzed using a Vanquish Flex binary UHPLC system (Thermo Scientific, Waltham, MA, USA) equipped with a diode array detector (DAD). Chromatographic separation was performed on an Acquity HSS T3 Column (100 × 2.1 mm, 1.8 μm particle size, 100Å pores; Waters) maintained at 30 °C. The mobile phase consisted of acetonitrile/formic acid (0.1%, *v*/*v*) (eluant A) and water/formic acid (0.1%, *v*/*v*) (eluant B) at a flow rate of 0.33 mL/min. The gradient elution program was as follows: 0–1 min at 85% B; 1–4 min, 85–70% B; 4–5 min, 70–50% B; 5–6.5 min, 50–40% B; 6.5–8.0 min, 40–1% B; 8.0–10 min, 1%B isocratic; 10–11 min, 1–85% B. The injected volume of the sample was 1 μL. The liquid chromatography system was coupled to an Exploris 120 Q-Orbitrap MS system (Thermo Scientific, Waltham, MA, USA). The mass spectrometer was operated with a heated electrospray ionization source in positive and negative ion modes. The key parameters were as follows: spray voltage, +3.5 and −3.5 kV; sheath-gas flow rate, 40 arbitrary units (arb. unit); auxiliary-gas flow rate, 10 arb. unit; sweep-gas flow rate, 1 arb. unit; capillary temperature, 360 °C; and auxiliary-gas-heater temperature, 300 °C. The scan modes were full MS with a resolution of 60,000 fwhm (at *m*/*z* 200) and ddMS2 with a resolution of 60,000 fwhm; the normalized collision energy was 30 V; and the scan range was *m*/*z* 85−1200. Internal mass calibration was operated using an EASY-IC internal calibration source, allowing single mass calibration for the full mass range. Data acquisition and processing were carried out with Xcalibur 4.5 and Free Style 1.7 (Thermo Scientific, Waltham, MA, USA), respectively.

### 4.5. RNA Extraction, cDNA Synthesis and RT-qPCR Expression Analysis

Before extraction, seeds were removed, and the pulp and skin were ground into a fine powder in liquid nitrogen. Total RNA was extracted from 1.2g of previously ground tissue, according to Nonis et al. (2012) [115]. Total RNA was resuspended in 30 µL of nuclease-free water. Subsequently, DNase digestion was performed with the On-Column DNase Digestion Set (Sigma–Aldrich, (Burlington, VT, USA)). The sample’s quantification was determined with a Qubit 4 fluorometer (Thermo Fisher). The purity of the samples was measured using NanoDrop 2000 (Thermo Fisher). RNA integrity was confirmed by a 1% agarose gel.

cDNA was reverse transcribed from one µg of total RNA using the M-MLV enzyme (Promega) according to the manufacturer’s instructions following the procedure described by Nonis et al. (2012) [115]. 

Quantitative real-time PCR was then performed using 1.2 µL of primer mix, 3.8 µL of cDNA, and 5 µL of PowerUp SYBR Green Master Mix (Applied Biosystems (Foster City, CA, USA)). Samples were amplified following the manufacturer’s instructions, and fluorescence was monitored with the step-one plus real-time PCR system (applied biosystems). Three technical replicates were carried out for each biological replicate. The primer sequences of the marker genes are reported in Table 2.

The automated Excel spreadsheet Q-Gene and modifications to the delta Ct method were used to perform the final calculations [117]. The housekeeping genes *VvTCPB* and *VvAIG1* were used to normalize gene expression [116]. Then, using equation 2 from the Q-Gene spreadsheet, levels of expression were calculated and expressed as arbitrary units of mean normalized expression.

### 4.6. RNA-Seq Analysis and Data Processing

Three biological replicates for each time point an experimental thesis were used for global transcriptomic analyses. Illumina directional sequencing of mRNA was performed at the Centro di Ricerca Interdipartimentale per le Biotecnologie Innovative (CRIBI—University of Padova, Italy) on a NovaSeq 6000 instrument (Illumina, San Diego, CA, USA). For each sample x replicate combination, 25–35 M paired-end reads of 150 nucleotides were generated. The quality of the reads was assessed using FastQC. The sequenced reads were pre-processed for low-quality sequence filter and adapter trimming with ERNEFILTER 2.1.1 [118] and Trimmomatic [119], respectively. High-quality reads were mapped to the *Vitis vinifera* PN40024.v4 genome obtained from the Ensembl Plants database using the spliced aligner HISAT2 [120]. Gene expression counts were generated using FeatureCounts software [121]. The differential expression analysis was carried out using DESeq2 [122], applying both the default Wald test for pairwise comparisons and the likelihood ratio test (LRT) for time-course analyses. Gene ontology enrichment analysis was performed using g:Profiler (https://biit.cs.ut.ee/gprofiler/gost (accessed on 28 November 2022)) while specific pathways were determined with KEGG Mapper (https://www.genome.jp/kegg/ (accessed on 28 November 2022)). Heatmaps were built based on information collected from MapMan software.

### 4.7. Data Treatment

Figures were prepared with Microsoft Excel, while statistical significance was determined using R software, version 4.0.3. Venn diagrams were built using Interactive Venn (http://www.interactivenn.net/ (accessed on 29 November 2022) and heatmaps were developed using Morpheus (https://software.broadinstitute.org/morpheus/ (accessed on 29 November 2022). 

## 5. Conclusions

This manuscript provides an in-depth, comprehensive overview of the biochemical and molecular effects of inactivated dry yeast extracts (DYE) on white grapes (var. Sauvignon Blanc). Our results demonstrated that DYE treatment did not interfere with the overall berry ripening process, as it did not affect sugars or acidity. Its action was, instead, more specifically and effectively directed towards selected metabolic and hormonal pathways. Specific aroma precursors, especially those related to sulphur and GSH metabolism, cys-3MH and GSH-3MH, typically contributing to Sauvignon Blanc aroma, were found at higher levels in grapes from vines sprayed with the DYE. Furthermore, the complete profile of amino acids as well as the RNA-seq transcriptional profiles strongly indicated that the treatment with DYE was able to trigger an unequivocal “priming” effect on berries by affecting the ROS balance and reducing oxidative damage, thus improving the stress tolerance, particularly heat stress. The DYE also stimulated a subset of ethylene and JA responses while interfering with some ABA and auxin-related genes, an aspect that will need further characterization for its functional relationships with the activation of stress responses exerted by the DYE.

## Figures and Tables

**Figure 1 plants-12-03423-f001:**
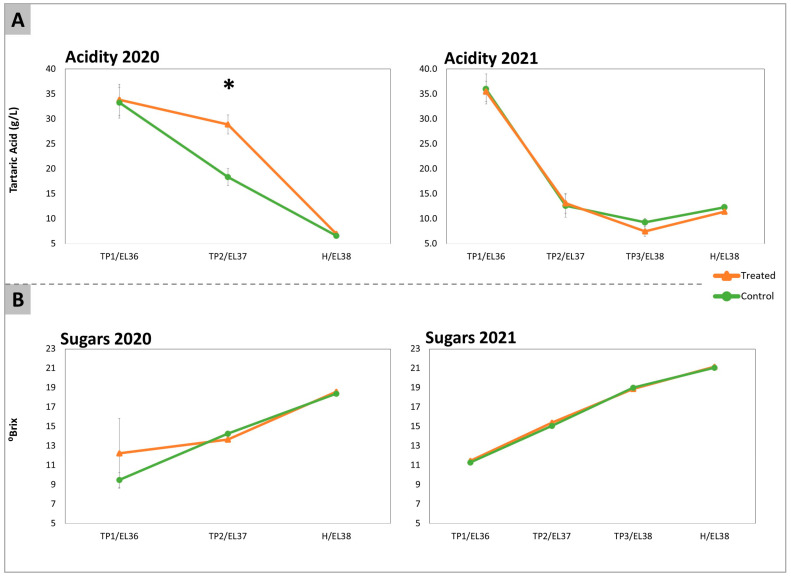
Total acidity (**A**) expressed in g/L of tartaric acid and total sugars (**B**) expressed in °Brix, determined in the samples of Sauvignon Blanc berries in the years 2020 and 2021 that had been collected from treated (orange lines) and control untreated (green lines) vines. Sugars were measured for each experimental condition individually on 60 berries belonging to each replica/treatment at each sampling date. The EL stages at each time point (TP1, TP2, and Harvest, H) were identified on the basis of the average sugar levels according to the modified EL system described by Coombe (1995) [16] as EL36, 37, and 38, respectively. Acidity was determined by titration of the juice obtained from 20 randomly collected berries. The significant differences (*) were determined with a *t*-test with a *p*-value < 0.05.

**Figure 2 plants-12-03423-f002:**
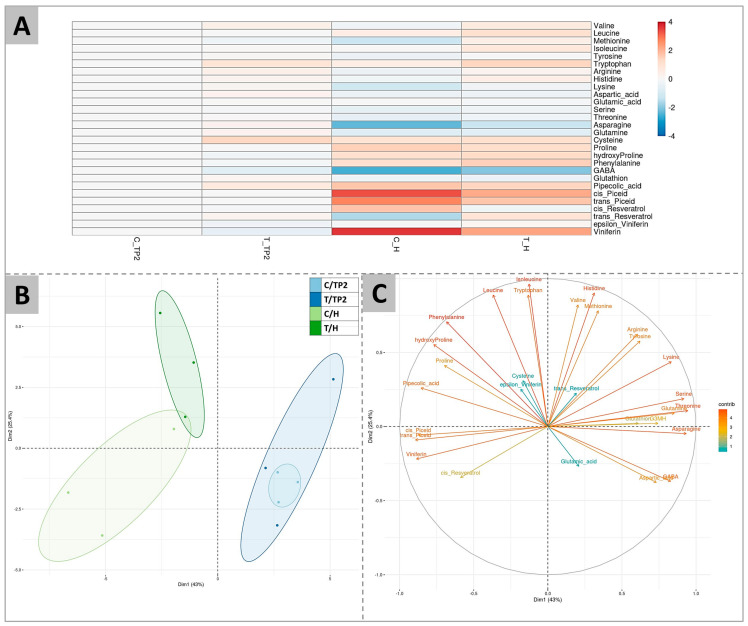
Amino acids and secondary metabolites profiling of berries at TP2 (EL37) and H (EL38) in 2020 and 2021 in control (C) and treated (T) samples. (**A**) is a heatmap of the log2 ratios of metabolite contents compared to control samples at TP2 (C_TP2), indicated by shades of red or blue according to the scale bar. (**B**) is a PCA showing the separation of samples according to ripeness (H vs. TP2) (x axis, dimension 1, explaining 43% of variability) or to the DYE treatment (y axis, dimension 2, explaining 25.4% of the variability). (**C**) shows the variable correlation plot of the PCA.

**Figure 3 plants-12-03423-f003:**
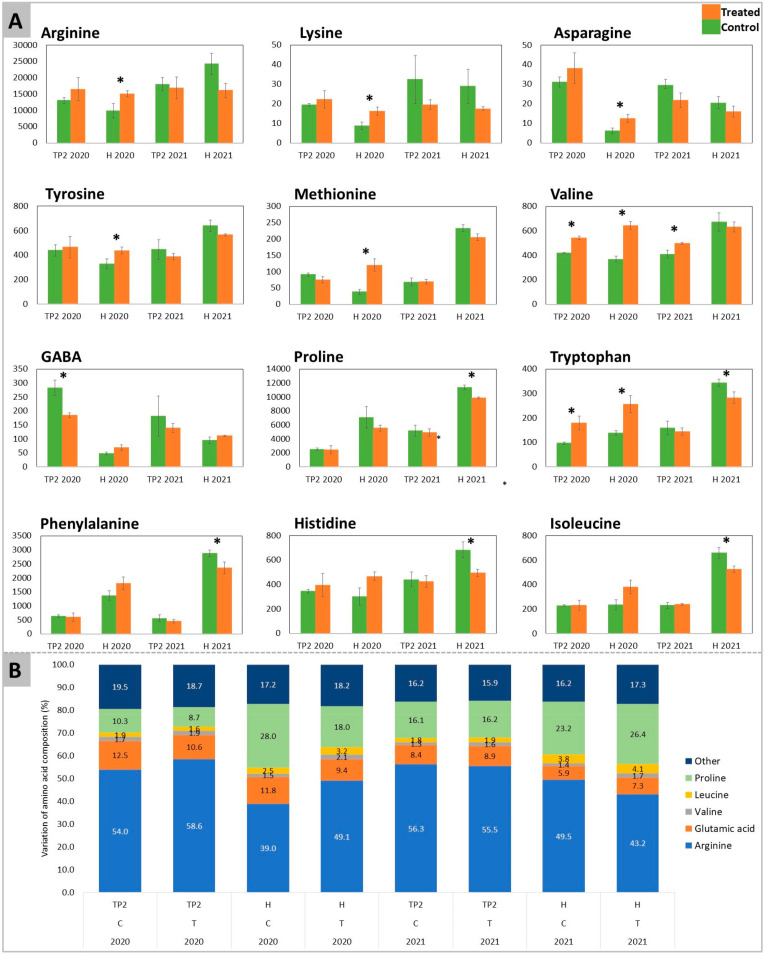
Relative quantitation of amino acids and GABA determined in Sauvignon Blanc’s berries for both years 2020 and 2021. In (**A**) the amino acids and GABA show significant differences between DYE-treated (orange) and untreated control (green) fruits (numbers represent the area of the peak). These compounds were determined in both years at time-point 2 (EL37) (after the second treatment) and at harvest. The significant differences (*) were determined with a *t*-test with a *p*-value < 0.05. (**B**) reports the variation (%) of amino acid composition throughout the experiment.

**Figure 4 plants-12-03423-f004:**
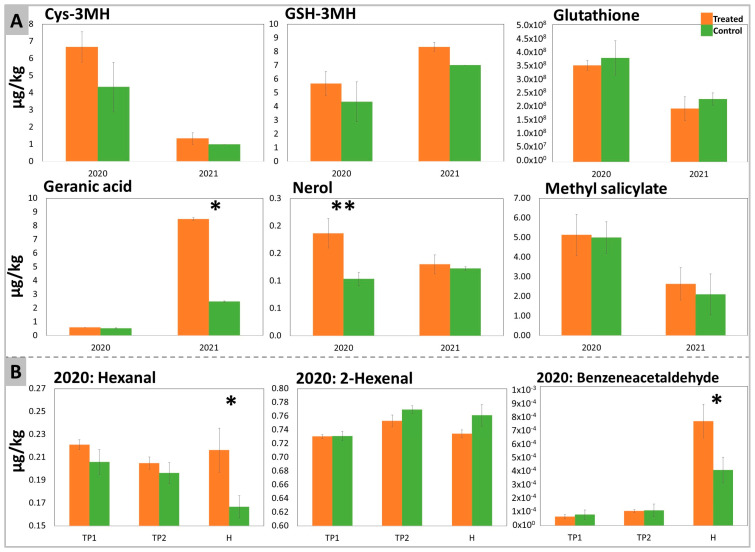
Aroma precursors determined in DYE-treated and untreated control Sauvignon Blanc’s berries. (**A**) shows bound aroma precursors and glutathione quantitated at harvest in the two years 2020 and 2021, while (**B**) represents free aroma precursors determined at all time points in samples collected in 2020. The significant differences were determined with a *t*-test with *p*-value < 0.1 (*) and with *p*-value < 0.05 (**).

**Figure 5 plants-12-03423-f005:**
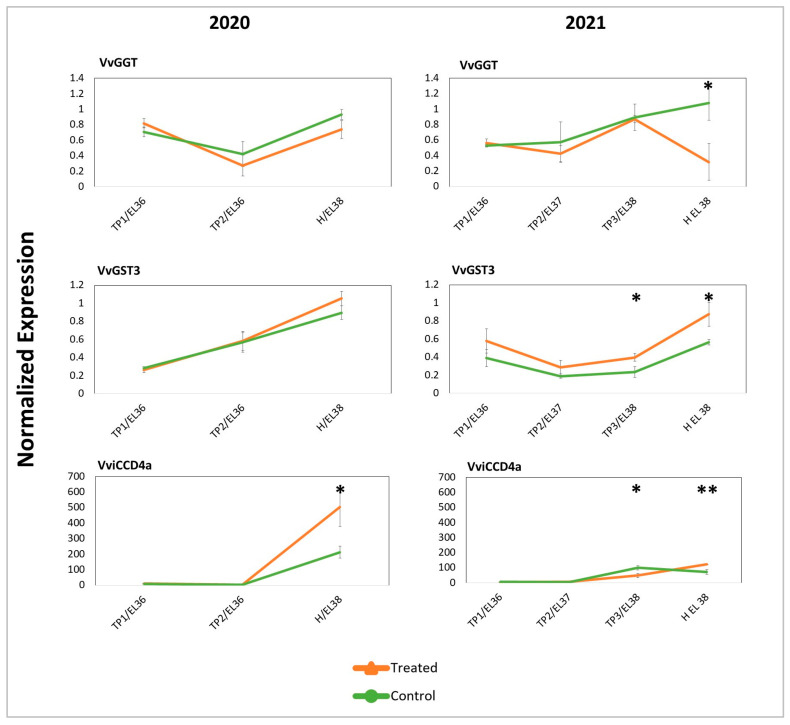
Normalized expression of genes encoding enzymes involved in the biosynthesis of berry aromas or of aroma precursors. The gene expression was evaluated in both years in treated and untreated control samples at subsequent developmental ripening stages (EL-36 to EL-38) and time points. *VvGGT* (Gamma-Glutamyl transferase) and *VvGST* (Glutathione S-transferase) are involved in glutathione metabolism, while *VviCCD4a* (Carotenoid Cleavage Dioxygenase) is involved in carotenoid metabolism and in the generation of norisoprenoids. Significant differences (*) (*p*-value < 0.1) and very significant differences (**) (*p*-value < 0.05) were determined with a *t*-test.

**Figure 6 plants-12-03423-f006:**
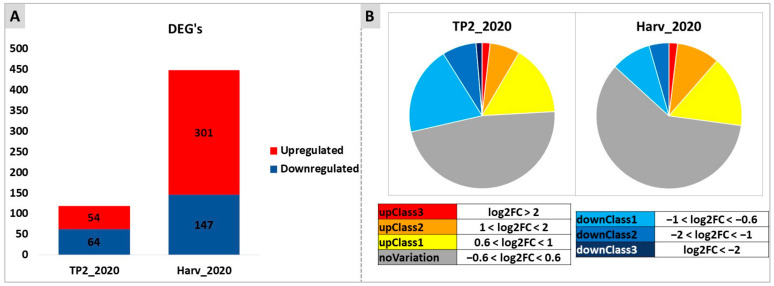
Overview of RNA-seq results of samples collected at TP2 and H in 2020. (**A**) symbolizes the differentially expressed genes (DEGs) between treated and control samples, observed at each time point (TP2 or H) separately. The two colors represent downregulated (blue) and upregulated (red) genes. (**B**) categorizes the DEGs to differentiate between highly differentially expressed (red and dark blue) (−2 > log2FC > 2) and slightly differentially expressed (yellow and light blue) (0.6 < log2FC < 1 and −1 < log2FC < −0.6, respectively) genes identified amongst the significant differential expressions.

**Figure 7 plants-12-03423-f007:**
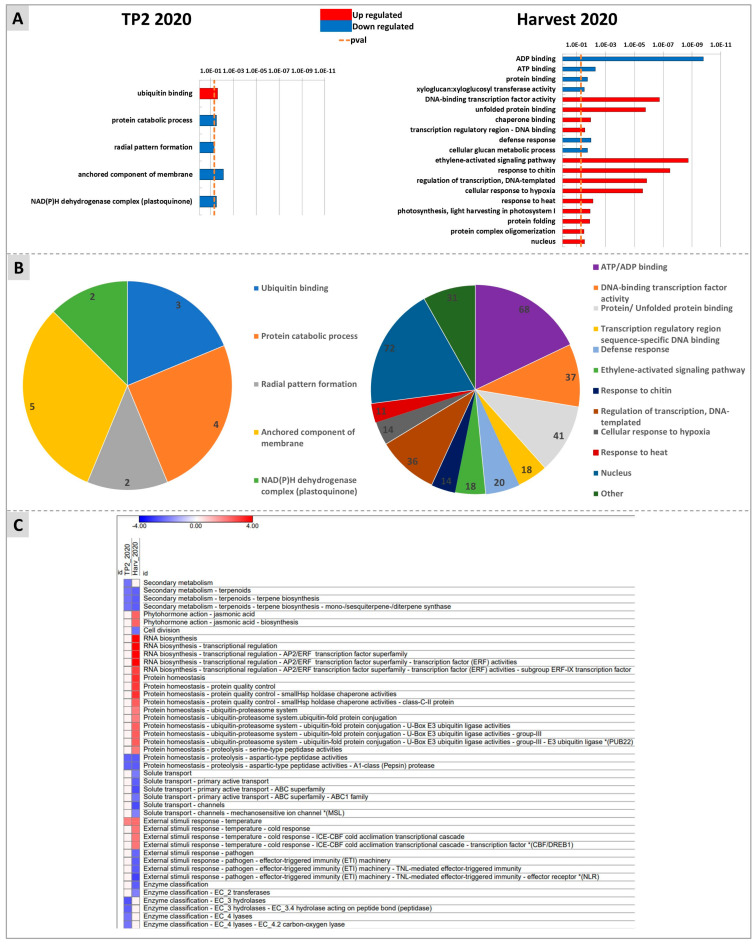
Enrichment analysis obtained with G:Profiler (**A**,**B**) and Mapman (**C**) softwares. (**A**) shows upregulated and downregulated pathways for TP2 (left panel) and H (right panel). (**B**) represents the number of DEGs involved in each pathway at TP2 (left panel) and H (right panel). (**C**) reports in a heatmap the pathways identified by Mapman as up- (red squares) or down-regulated (blue squares) at TP2 or at H.

**Figure 8 plants-12-03423-f008:**
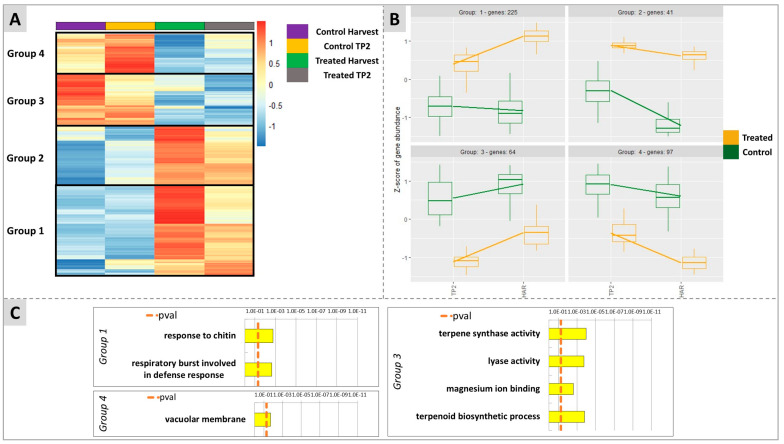
Results of the untargeted analysis considering the interaction of time per treatment on the modulation of ripening-related genes in the berry. (**A**) is a heatmap showing the clustering of four groups according to gene’s behavior throughout ripening from TP2 to H. (**B**) shows the dynamics of expression of each group of genes indicated in the corresponding clusters identified in (**A**). (**C**) reports the enrichment analysis made with G:Profiler for each group. Group 2 is not shown since no enrichment could be identified due to the low number of genes for this group.

**Figure 9 plants-12-03423-f009:**
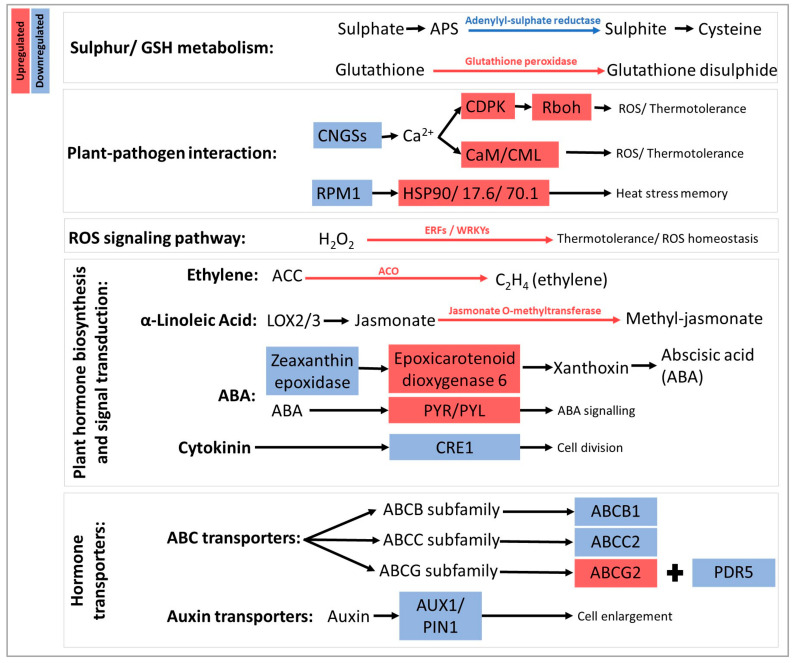
Schematic overview of the transcriptional effects of DYE treatment during berry ripening, determined with KEGG analysis. Highlighting of genes in blue and blue arrows indicates downregulation, while genes and arrows marked in red indicate upregulation. Sulphur/glutathione (GSH) metabolism: APS, adenosine 5′-phosphosulfate. Plant-pathogen interactions: CNGSs, cyclic nucleotide gated channels; RPM1, disease resistance protein 1; CDPK, calcium-dependent protein kinase; Rboh: respiratory burst oxidase homolog; CaM/CML, calmodulin and calmodulin-like; HSP, heat shock proteins. ROS signaling pathway: WRKYs, workies; ERFs; ETS2 repressor factors. Plant hormone biosynthesis and signal transduction: ACO, 1-aminocyclopropane-1-carboxylic acid oxidase; ACC, 1-aminocyclopropane-1-carboxylic acid; LOX, lipoxygenase; ABA, abscisic acid; PYR/PYL, Pyrabactin resistance/Pyrabactin resistance-like; CRE1, cytokinin response 1. Hormone transporters: AUX1, auxin transporter 1; PIN1, auxin efflux carrier.

**Figure 10 plants-12-03423-f010:**
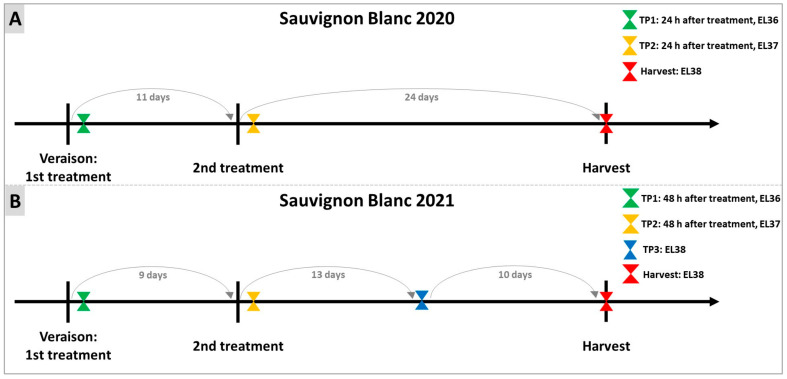
Experimental design of treatments of Sauvignon Blanc with DYE and sampling for the two years 2020 (**A**) and 2021 (**B**). On both years, Sauvignon Blanc vineyards were sprayed with the DYE at the beginning of veraison and again after a period of 9–11 days from the first treatment. Sampling was performed 24/48 h after each treatment (TP1 (EL36, 10.1–14 ⁰Brix) and TP2 (EL37, 14.1–18 ⁰Brix)) and at H (EL38, 18.1–22 ⁰Brix). In the year 2021, an additional sampling, TP3 (EL38, 18.1–22 ⁰Brix), was performed between TP2 and H.

**Table 1 plants-12-03423-t001:** Genes differently expressed in the analysis of the combined effects of time per treatment with DYE. The columns in the table report the gene ID in *Vitis vinifera*, the closest corresponding Arabidopsis homologue and its biochemical function identified in Arabidopsis, the group into which they clustered (groups of Figure 8), and their putative biological role as identified by the literature.

Grapevine Gene	Arabidopsis Homologue	Function in Arabidopsis	Group	Reference	Putative Role
Response to chitin and regulatory burst involved in defence response					
*Vitvi08g02053*	*AT2G35930*	Plant U-box 23	1	Cho et al., 2008 [22] Trujillo et al., 2008 [23]	Ubiquitin ligase homologous to U-Box E3 playing a role in response to drought stress.
*Vitvi08g02051*	*AT2G35930*	Plant U-box 23	1	Cho et al., 2008 [22] Trujillo et al., 2009 [23]	Ubiquitin ligase homologous to U-Box E3 playing a role in response to drought stress.
*Vitvi12g00139*	*AT1G29340*	Plant U-box 17	4	Wang et al., 2006 [24]	In tobacco: ubiquitin-proteasome system in defences against pathogens.
*Vitvi15g01003*	*AT5G24110*	WRKY Transcription Factor Group III	1	Scarpeci et al., 2013 [25]	AtWRKY30, putative role in the activation of defence responses.
*Vitvi07g03110*	*AT5G65530*	Protein kinase superfamily protein	1	Reiner et al., 2015 [33]	RLCK VI, regulating basal resistance to powdery mildew.
ROS Homeostasis					
*Vitvi02g00332*	*AT1G63460*	Glutathione peroxidase 8	1	Gaber et al., 2012 [27]	Involvement in the suppression of oxidative damage.
*Vitvi03g00500*	*AT4G34410*	ERF109	1	Zhang et al., 2019 [34] Li et al., 2021 [35]	Wounding, JA and salt stress inducible; regulates ROS production and stress adaptation.
*Vitvi13g00097*	*AT5G58530*	Glutathione oxidoreductase	2	No reference	Reduction of thiol groups in proteins.
*Vitvi14g00183*	*AT1G09090*	RBohB	2	Wang et al., 2014 [28] Müller et al., 2009 [29]	ABA inducible in seed after ripening and involved in conferring thermotolerance.
Heat Shock Proteins					
*Vitvi13g00491*	*AT3G46230*	Heat shock protein 17.4	1	Sewelam et al., 2019 [36] McLoughlin et al., 2016 [37]	Response to ROS and several abiotic stresses, including cold and heat.
*Vitvi06g00443*	*AT5G02500*	Heat shock cognate protein 70.1	2	Tiwari et al., 2020 [38]	Negative regulator of basal heat tolerance.
*Vitvi08g02189*	*AT5G02500*	Heat shock cognate protein 70.1	1	Tiwari et al., 2020 [38]	Negative regulator of basal heat tolerance.
*Vitvi09g00045*	*AT1G54050*	HSP20-like chaperones superfamily	1	Lee and Bailey-Serres, 2019 [39]	Hypoxia related stress induction.
*Vitvi02g00025*	*AT5G52640*	Heat shock-like protein 90.1	2	Wang et al., 2016 [40]	Interacts with disease resistance signaling components and is required for RPS2-mediated resistance.
*Vitvi16g01103*	*AT5G52640*	Heat shock-like protein 90.1	2	Wang et al., 2016 [40]	Interacts with disease resistance signaling components and is required for RPS2-mediated resistance.
*Vitvi01g01846*	*AT2G03440*	Nodulin-related protein 1	1	Fu et al., 2010 [41]	Negative regulation of ABA response and abiotic stress (cold) inducible.
*Vitvi04g00092*	*AT2G26150*	Heat shock transcription factor A2	1	Friedrich et al., 2021 [30]	Regulating heat stress memory genes and thermotolerance.
*Vitvi07g00078*	*AT3G22830*	Heat shock transcription factor A6B	1	Huang et al., 2016 [31]	Positive mediator of ABA dependent thermotolerance and drought resistance.
*Vitvi01g00372*	*AT5G38110*	Anti- silencing function 1b	1	Lario et al., 2013 [32] Weng et al., 2014 [42]	A positive regulator of basal and acquired thermotolerance.
*Vitvi14g02461*	*AT2G47180*	Galactinol synthase 131	3	Jang et at., 2018 [43]	Role in improving oxidative stress tolerance by increasing galactinol biosynthesis in Arabidopsis.
Hormone biosynthesis and signalling—Lipid metabolism, JA biosynthesis and Response					
*Vitvi13g01780*	*AT3G45140*	Lipoxygenase 2	1	No reference	Lipoxygenase involved in JA biosynthesis.
*Vitvi09g00085*	*AT1G17420*	Lipoxygenase 3	1	Chávez-Martínez et al., 2020 [44] Yang et al., 2020 [45]	Lipoxygenase involved in JA biosynthesis.
*Vitvi04g02169*	*AT1G19640*	Jasmonic acid carboxyl methyltransferase	1	Kim et al., 2009 [46] Wu et al., 2008 [47]	Catalyses the formation of methyl jasmonate from JA.
*Vitvi09g00038*	*AT3G14225*	GDSL-motif lipase 4 hydrolase	1	Oh et al., 2005 [48]	Lipase involved in resistance to necrotrophic pathogens.
*Vitvi10g00669*	*AT5G45670*	GDSL-like Lipase/Acylhydrolase	1	No reference	Lipase. Unknown role.
*Vitvi07g00039*	*AT2G30550*	Alpha/beta-Hydrolases superfamily protein	1	Dervisi et al., 2020 [49]	Role in JA biosynthesis. Lipase hydrolysing phosphatidylcholine, glycolipids, triacylglycerol.
*Vitvi15g00298*	*AT3G03520*	Phospholipase C3	1	Krčková et al., 2015 [50]	Positive regulator of thermotolerance, induced by phosphate starvation and abiotic stresses.
*Vitvi13g00864*	*AT5G07010*	Sulfotransferase 2A	1	Gidda et al., 2003 [51]	Acts on 11- and 12-hydroxyjasmonic acid. Involved in reducing excess JA levels.
*Vitvi13g01379*	*AT5G07010*	Sulfotransferase 2A	3	Gidda et al., 2003 [51]	Acts on 11- and 12-hydroxyjasmonic acid. Involved in reducing excess JA levels.
*Vitvi14g01283*	*AT5G38710*	Proline Dehydrogenase	4	Funck et al., 2010 [52]	Upregulated during salt stress.
*Vitvi12g02545*	*AT4G08500*	MAPK/ERK kinase 1	4	Kong et al., 2012 [53]	Negative regulator of wound and immune responses.
*Vitvi11g01479*	*AT5G51350*	Leucine-rich repeat transmembrane protein kinase	4	Gursanscky et al., 2016 [54]	Represses genes associated with ethylene and JA.
Hormone biosynthesis and signalling—Ethylene Biosynthesis and signaling and hypoxic responses					
*Vitvi10g02409*	*AT1G05010*	Ethylene-forming enzyme ACO4	1	Moon et al., 2020 [55]	BR repressible ethylene biosynthetic gene involved in inhibition of growth.
*Vitvi16g01438*	*AT5G47230*	ERF5	1	Son et al., 2012 [56] Moffat et al., 2012 [57]	Involved in chitin-induced innate immunity and a positive regulator of JA/ethylene-responsive genes.
*Vitvi12g00274*	*AT5G44210*	ERF9	1	Maruyama et al., 2013 [58]	Regulator of plant defence against necrotrophic fungi mediated by the DEAR1-dependent ethylene/JA signaling pathway.
*Vitvi19g01784*	*AT3G15210*	ERF4	1	Yang et al., 2005 [59]	Modulates ethylene and ABA responses.
*Vitvi18g02240*	*AT1G71450*	Integrase-type DNA-binding superfamily	1	Chen et al., 2015 [60]	Negatively regulates ethylene response.
*Vitvi06g00667*	*AT2G30020*	AP2C1	1	Schweighofer et al., 2007 [61]	Negative regulator of ethylene and JA synthesis and responses of resistance to *Botrytis c.*
*Vitvi07g00357*	*AT2G47520*	ERF B-2 of ERF/AP2 transcription factor family	1	Licausi et al., 2010 [62]	Hypoxia responsive ERF.
*Vitvi08g00793*	*AT2G38470*	WRKY33	1	Birkenbihl et al., 2012 [63] Tang et al., 2021 [64]	WRKY transcription factor regulator of *Botrytis* resistance and hypoxia responses by direct regulation of RAP2.2.
*Vitvi15g01090*	*AT4G01250*	WRKY22	1	Hsu et al., 2013 [65]	WRKY22 family transcription factor, hypoxia inducible and regulating innate immunity in hypoxia.
*Vitvi16g01444*	*AT5G51190*	ERF105	1	Bolt et al., 2017 [66]	Transcription factor required for freezing tolerance and cold acclimation.
*Vitvi17g00787*	*AT1G60190*	ERF105	1	Bolt et al., 2017 [66]	Transcription factor required for freezing tolerance and cold acclimation.
*Vitvi16g01434*	*AT1G60190*	ERF105	1	Bolt et al., 2017 [66]	Transcription factor required for freezing tolerance and cold acclimation.
*Vitvi16g01432*	*AT1G60190*	ERF105	1	Bolt et al., 2017 [66]	Transcription factor required for freezing tolerance and cold acclimation.
*Vitvi18g00295*	*AT1G60190*	ERF105	1	Bolt et al., 2017 [66]	Transcription factor required for freezing tolerance and cold acclimation.
*Vitvi16g01423*	*AT1G60190*	ERF105	1	Bolt et al., 2017 [66]	Transcription factor required for freezing tolerance and cold acclimation.
*Vitvi02g00407*	*AT1G63030*	Integrase-type DNA-binding superfamily	1	Lehti-Shiu et al., 2015 [67]	Negative regulator of gibberellic acid biosynthesis.
Hormone biosynthesis and signalling—ABA biosynthesis and signaling					
*Vitvi17g00116*	*AT3G48530*	SNF1-related protein kinase regulatory subunit gamma 1	1	Van Dingenen et al., 2019 [68]	Regulates responses to sugar availability, negative regulator of HXK1.
*Vitvi02g00114*	*AT4G11070*	WRKY family transcription factor	1	Ding et al., 2014 [69]	Regulation of ABI3 independent from ABA, positive regulating ABA sensitivity.
*Vitvi13g00058*	*AT2G40140*	Zinc finger (CCCH-type) family protein	1	AbuQamar et al., 2006 [70]	Positive regulator of *Botrytis* resistance and negative regulator of ABA.
*Vitvi01g00956*	*AT2G02820*	MYB domain protein 88	1	Xie et al., 2010 [71]	Sensing and/or transducing drought and saline stress.
*Vitvi14g01499*	*AT5G13180*	NAC domain containing protein 83	1	Yang et al., 2011 [72]	Molecular link integrating plant responses to environmental stresses and leaf longevity.
*Vitvi10g00063*	*AT1G62300*	WRKY family transcription factor	1	Chen et al., 2009 [73]	Regulates the ABA responsive gene Phosphate1 (Pho1) expression in response to low phosphate.
*Vitvi08g00793*	*AT2G38470*	WRKY transcription factor family.	1	Liu et al., 2015 [26]	Negative regulator of ABA signalling and positive regulator of resistance to *Botrytis*.
*Vitvi08g00298*	*AT5G04870*	CDPK1	1	Yu et al., 2007 [74]	Positive regulator of ABA responses in grape berry.
*Vitvi15g01084*	*AT2G46140*	Late embryosis abundant protein	1	Candat et al., 2014 [75]	Involved in protection from abiotic stress: desiccation and cold.
*Vitvi16g00941*	*AT4G25480*	DREB subfamily A-1 of ERF/AP2 transcription factor family (CBF3)	1	Kasuga et al., 1999 [76]	Involved in conferring resistance to drought and freezing stress.
*Vitvi02g01288*	*AT4G19170*	9-cis-epoxycarotenoid dioxygenase 4	2	Bruno et al., 2016 [77]	Production of acyclic regulatory metabolites.
*Vitvi05g00548*	*AT3G51895*	Sulphate transporter 3;1	1	Cao et al., 2014 [78]	Co-regulation of S-metabolism and ABA biosynthesis.
*Vitvi18g03100*	*AT5G47550*	Cystatin/monellin superfamily protein	1	Song et al., 2017 [79]	Positive role in the heat shock-responsive expression of AtCYS5.
*Vitvi01g00353*	*AT1G18390*	Serine/Threonine kinase	2	Lim et al., 2015 [80]	Involved in ABA-mediated signaling and drought resistance.
*Vitvi18g01703*	*AT1G71960*	ATP-binding cassette family G25 ABCG25	3	Kuromori et al. 2021 [81]	Involved in intercellular ABA import inside cells.
*Vitvi04g00423*	*AT1G15520*	ATP-binding cassette family G40 ABCG40	3	Kang et al., 2015 [82]	Necessary for ABA export from cells.
*Vitvi05g00963*	*AT3G24220*	9-cis-epoxycarotenoid dioxygenase 6	3	Seo et al., 2006 [83]	ABA biosynthetic gene.
*Vitvi17g00601*	*AT1G74670*	GASA6	3	Qu et al., 2016 [84]	ABA repressible and GA inducible; cell wall regulator.
*Vitvi12g00685*	*AT5G35750*	Histidine kinase 2	4	Wulfetange et al., 2011 [85]	Negative regulator of ABA sensitivity in stomata.
Hormone biosynthesis and signalling—Auxin and CK Response					
*Vitvi11g00394*	*AT3G23050*	Indole-3-acetic acid 7	2	Cui et al., 2013 [86]	AUX/IAA family factor. Negative regulator of pathogenicity, through regulation of auxin response.
*Vitvi12g00594*	*AT4G27280*	Calcium-binding EF-hand family protein	1	Hazak et al., 2019 [87]	Ca^2+^ dependent transducer of auxin-regulated gene expression.
*Vitvi16g01201*	*AT4G27950*	Cytokinin response factor 4	1	Zwack et al., 2016 [88]	Cytokinin response factor 4 (CRF4) is a positive regulator of freezing tolerance
*Vitvi17g00210*	*AT1G73590*	PIN1 auxin efflux carrier	3	Banasiak et al., 2019 [89]	Auxin efflux transporter.
*Vitvi13g00019*	*AT2G38120*	AUX1	4	Bennet et al., 1996 [90]	Auxin influx transporter.
*Vitvi05g00476*	*AT5G19530*	Spermine synthase	4	Hanzawa et al., 2003 [91]	Biosynthesis of polyamine spermine, promoting growth of organs.
*Vitvi17g00157*	*AT2G01940*	C_2_H_2_-like zinc finger protein	4	Morita et al., 2006 [92]	Transcription factor involved in shoot gravitropism and gravity perception.

**Table 2 plants-12-03423-t002:** Sequences of primers used for targeted real-time PCR expression analysis. *VvTCPB* and *VvAIG1* are the reference genes [116]; *VvGGT* (Gamma-Glutamyltransferase) [12] and *VvGST* (Glutathione S-transferase) [12] are involved in glutathione metabolism; *VviCCD4a* (Carotenoid Cleavage Dioxygenase) [1] is a gene putatively involved in carotenoid metabolism and generation of norisoprenoids; *VviLinNer1* is a Linalool synthase [19].

Gene	Primer	Primer F	Primer R	Reference
*VvTCPB* *(Vitvi18g00138)*	TCPB	CAGACAGTGATTGACAGCCGAGTT	ATCCCTGCGTGGCTTTCTTCC	[116]
*VvAIG1* *(Vitvi18g02131)*	AIG1	GGAAGATTATTTGGGCCGTGAG	ACTTCTTGGCTTCATCCTTGGTC	[116]
*VviLinNer1 (Vitvi10g02128)*	Ner1	GTGGGCGAGTTTATGCAACG	CCCTGAACTAACGGCCCCAT	[19]
*VviCCD4a (Vitvi02g01286)*	CCD4a	CGAGGCATCCGCTATCCACA	CACGTCCAGCTTCACCACTCC	[1]
*VvGGT (Vitvi11g00234)*	GGT	TGGCAACAGCTTAGAGGCAGTA	CCCACCTGCCTTTCTCACAT	[12]
*VvGST3* *(Vitvi17g01467)*	GST3	TGCAAAGGTGTTGGACATCTATG	TGTGAATGGAAGGTGGCTAAGA	[12]

## Data Availability

The datasets generated during and/or analyzed during the current study are listed in the text and its additional files.

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
