# Peer review of "Metabolic and Molecular Rearrangements of Sauvignon Blanc (Vitis vinifera L.) Berries in Response to Foliar Applications of Specific Dry Yeast"

_plants, 2023, doi:10.3390/plants12193423_

Round 1
Reviewer 1 Report
In this work aimed at unravelling the metabolic and molecular responses of Sauvignon 18 Blanc berries to DYE treatments, for two consecutive years. Grapes were analysed for sugars, acidity, free and bound aroma precursors, amino acids, and targeted and untargeted RNA-Seq transcriptional profiles. The results obtained indicated that the treatment did not interfere with the technological ripening aspects (sugars and acidity). The manuscript show important phases of the treatment.
In references section all references presented dos times same number?
Author Response
In references section all references presented dos times same number?
Reply: We have changed the doubled number present in the references, according to the reviewer’s comment. We do not know how this may have happened and we apologize for the inconvenience.
Reviewer 2 Report
Language needs editing
Author Response
Does the introduction provide sufficient background and include all relevant references? Can be improved.
Reply: we thank the reviewer for his/her constructive comment. We have added more background, namely regarding aroma compounds (64-69) and stress mitigation strategies and their limitations (81-88) in the revised text. We have included comments on the lack of information regarding the molecular effect of DYE and also comments on the novelty of our work, linking it to the current knowledge in the field. We have also checked the text again to improve the clarity. We hope that in the present version the revised introduction will meet the reviewers requests.
Language needs editing.
Reply: we believe we have improved the language, besides checking the text several times for typos etc (please see all changes highlighted in the text), we have tried to make the sentences more fluent by simplifying some sentences and by dividing some other long sentences into 2 separate sentences to make them clearer (e.g., 97-100, 596-601, 644-650). We are confident that the new text is significantly improved compared to the original one and hope that it will be satisfactory for the reviewer.
Reviewer 3 Report
Dear Authors,
The present study evaluates the metabolic and molecular responses of Sauvignon Blanc berries to dry yeast extract treatments. The research subject is interesting and brings scientific important data in the field. Some changes of the manuscript should nevertheless be performed in order to improve its quality. Following specific changes should thus be performed:
Major changes
Introduction: Authors need to clarify what their study brings in novelty, compared to similar studies found in scientific literature, that are presented. It is very important to state what exactly you bring in novelty in order to express your originality. Please avoid formulations as “several studies”, they bring a lot of uncertainty. In the last paragraph, authors do not need to refer to materials, methods or results, it is enough to clearly present the purposes of the study.
Results and Discussions: Novelty and originality of the study is essential to be emphasized in this section once again. Authors need to emphasize this in terms of results, not purposes, as the Introduction should. Please specify where it’s the case.
Materials and Methods: You do not offer references for all of the assays. Are all methods completely new? Because, if they are, you need to offer them a different approach.
References need to follow the recommendations of the journal in terms of editing.
All these suggested changes should be performed in order to bring further improvements to the manuscript.
Quality of language is fine, only minor corrections are needed.
Author Response
Does the introduction provide sufficient background and include all relevant references? Can be improved.
Reply: we thank the reviewer for his/her constructive comment. We have added more background, namely regarding aroma compounds (64-69) and stress mitigation strategies and their limitations (81-88) in the revised text. We have additionally included comments on the lack of information regarding the molecular effects of DYE (107-110) and comments on the novelty of our work and linked these comments to the current knowledge in the field. The novelty has been highlighted in the final part of the introduction (122-127). We have also checked the text again to improve the clarity. We hope that in the present version the revised introduction will meet the reviewer’s requests.
Are the methods adequately described? Can be improved.
Reply: we have added more details to section 4.3.1 (832-848) and 4.3.2 (857-898) and a reference to section 4.3.3 (904-905). The text has been thoroughly checked again.
Introduction: Authors need to clarify what their study brings in novelty, compared to similar studies found in scientific literature, that are presented. It is very important to state what exactly you bring in novelty in order to express your originality.
Reply: we agree with the reviewer’s constructive comment and we appreciate his/her constructive comment. We understand that the novelty of our work was not highlighted clearly enough in the original version of the manuscript. We have now improved the abstract and the paragraph of the aims of the study in the final part of the introduction(119-127), by stating that our study not only focused on the molecular mechanisms of aroma improvement but also, we were able to provide for the first time a comprehensive overview (by RNA-Seq) of the transcriptional changes induced by DYE treatment in grapevine berries. This comprehensive overview was missing, as we have underlined, we hope clearly now, in the introduction. We hope that we have now better highlighted and explained in the introduction that this systematic approach has led us to identify abiotic stress tolerance pathways that might explain the elicitor-like response of DYE and their ability to help to counteract climate change adverse effects and we have identified some hormonal pathways to be part of it, an aspect that has never been shown by previous studies.
Please avoid formulations as “several studies”, they bring a lot of uncertainty. In the last paragraph, authors do not need to refer to materials, methods, or results, it is enough to clearly present the purposes of the study.
Reply: we have changed the expression “several studies” for “previous studies” (line 93) and removed the materials, methods, and results from the aims paragraph, as requested by the reviewer.
Results and Discussions: Novelty and originality of the study is essential to be emphasized in this section once again. Authors need to emphasize this in terms of results, not purposes, as the Introduction should. Please specify where it’s the case.
Reply: As done for the introduction, we have underlined the novel findings of the work more clearly in the new version of the discussion at the end of some paragraphs (601-603, 659-662, 697-702, 713-715) and at the end of the discussion (763-774). We hope that the reviewer will find the new text more effective in bringing the message.
Materials and Methods: You do not offer references for all of the assays. Are all methods completely new? Because, if they are, you need to offer them a different approach.
Reply: We have added the reference in the amino acid assay (904-905).
References need to follow the recommendations of the journal in terms of editing.
Reply: we have removed the doubled numbers in the references and corrected some imprecisions in the references along the text (lines 100, 102, 560, 561 and 570).
All these suggested changes should be performed in order to bring further improvements to the manuscript.
Quality of language is fine, only minor corrections are needed.
Reply: we have thoroughly revised the manuscript and corrected some typos and simplified some sentences by shortening them or dividing them into 2 separate sentences (97-100, 596-601, 644-650). We think that in this way the text will be clearer and more readable.
Reviewer 4 Report
The work presents a well-conducted experimental plan, with analyses relevant to the intended purpose. It is a robust study, and in my opinion, suitable for publication. Some points need to be improved:
L504, 506, 515 and throughout the manuscript: Standardize the citation form according to the journal standard.
L684: put Vitis vinifera L. in italics.
Sections 4.3.1 and 4.3.2: Provide more details of the analytical conditions: instrument, chromatographic conditions, how the compounds were identified and quantified. In the same way as described in section 4.4
Section 4.4: It is mentioned that the extraction was from grape powder, but the protocol for obtaining the powder was not described.
Author Response
We thank the reviewer for his/her constructive comments and suggestions. We report below our replies to his/her specific comments
L504, 506, 515 and throughout the manuscript: Standardize the citation form according to the journal standard.
Reply: corrected some imprecisions in the references along the text (lines 100, 102, 560, 561 and 570).
L684: put Vitis vinifera L. in italics.
Reply: corrected (line 779), as requested. We have also changed the names of the genes into italic, we apologise for the mistakes.
Sections 4.3.1 and 4.3.2: Provide more details of the analytical conditions: instrument, chromatographic conditions, how the compounds were identified and quantified. In the same way as described in section 4.4.
Reply: we have added more details to section 4.3.1 832-848) and 4.3.2 (857-898) regarding instrument, chromatographic conditions, extraction method, and others, as requested by the reviewer.
Section 4.4: It is mentioned that the extraction was from grape powder, but the protocol for obtaining the powder was not described.
Reply: we have added a brief description of the methodology adopted for the preparation of powder (905-907). We hope we have correctly interpreted the request of the reviewer.
Round 2
Reviewer 3 Report
Dear Authors,
The present study evaluates the metabolic and molecular responses of Sauvignon Blanc berries to dry yeast extract treatments. The authors performed most of the suggested changes after the first sound of review. Following specific changes should still be performed:
Major changes
Introduction: I did not find novelty and originality of the study clearly stated.
References need to follow the recommendations of the journal in terms of editing.
All these suggested changes should be performed in order to bring further improvements to the manuscript.
English is fine, minor changes in language are required.
Author Response
We sincerely thank the reviewer for the rapid revision of our manuscript. References need to follow the recommendations of the journal in terms of editing. Reply: We apologise for the incorrect formatting of the references and we have now formatted them according to the journal guidelines, as suggested. Introduction: I did not find novelty and originality of the study clearly stated. Reply: In our revised version of the manuscript, in the lines 107 to 127, we have explained that our manuscript provides a first systematic analysis of the molecular signalling and mode of action of dry yeast extract (DYE) treatment to grapes, by adopting an RNA-Seq approach combined with metabolic analyses. In the same paragraph we have also explained that this information is completely lacking in the current literature and that our work fills this gap. In our opinion this information rather clearly indicates the novelty of the work and the new information that it provides. Perhaps the reviewer 3 may have not read this part due to problems in tracking the revisions? Or, alternatively, she/he asks for an even more explicit statement ? Nevertheless, we find that perhaps this latter option may not be necessary since the above mentioned lines indicate what our work has added to the current knowledge.